# Personalized Image Editing in Text-to-Image Diffusion Models via Collaborative Direct Preference Optimization

**Connor Dunlop**       **Matthew Zheng**[*]       **Kavana Venkatesh**[*]       **Pinar Yanardag**
Virginia Tech, Blacksburg, VA
`{cdunlop, pinary}@vt.edu`
`http://personalized-editing.github.io`

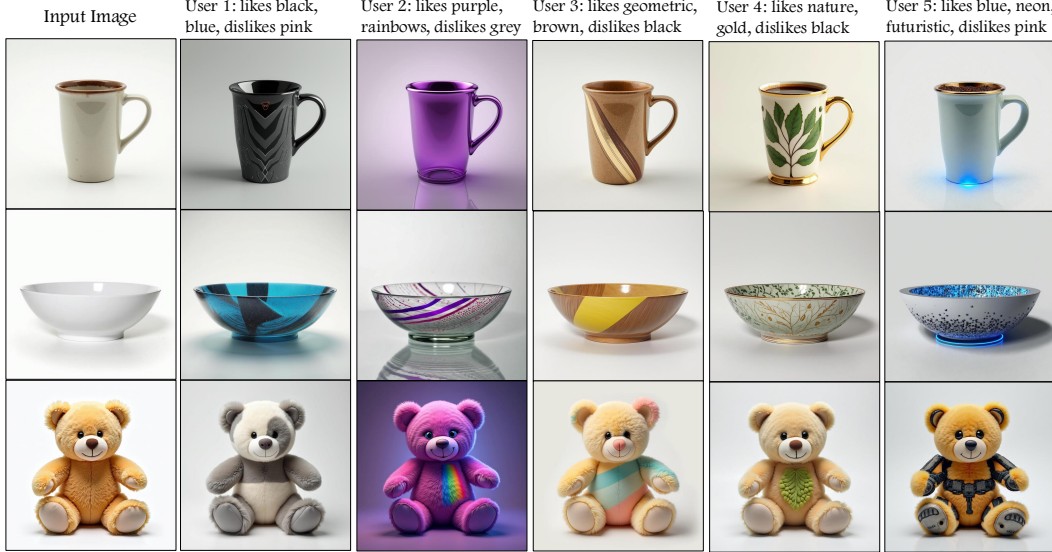

Figure 1: Our framework performs *personalized* image editing aligned with user's preference via a novel DPO objective that learns user preferences while leveraging collaborative signals from like-minded individuals.

## Abstract

Text-to-image (T2I) diffusion models have made remarkable strides in generating and editing high-fidelity images from text. Yet, these models remain fundamentally generic, failing to adapt to the nuanced aesthetic preferences of individual users. In this work, we present the first framework for personalized image editing in diffusion models, introducing Collaborative Direct Preference Optimization (C-DPO), a novel method that aligns image edits with user-specific preferences while leveraging collaborative signals from like-minded individuals. Our approach encodes each user as a node in a dynamic preference graph and learns embeddings via a lightweight graph neural network, enabling information sharing across users with overlapping visual tastes. We enhance a diffusion model's editing capabilities by integrating these personalized embeddings into a novel DPO objective, which jointly optimizes for individual alignment and neighborhood coherence. Comprehensive experiments, including user studies and quantitative benchmarks, demonstrate that our method consistently outperforms baselines in generating edits that are aligned with user preferences.

---

[*]Equal contribution.

39th Conference on Neural Information Processing Systems (NeurIPS 2025).

# 1   Introduction

Text-to-image (T2I) diffusion models have achieved remarkable success in visual content generation, enabling high-quality image synthesis from textual descriptions [32, 31, 18]. Building on this success, recent advancements have begun to repurpose diffusion models for image editing – allowing users to modify a given image via text prompts or other guidance [42, 5]. Despite progress, achieving a desired edit often demands substantial manual effort. Existing editing approaches do not account for the individual user's preferences. They treat image editing as a one-size-fits-all problem: the model does not adapt to a particular user's style, and it assumes the same definition of a 'good' edit for everyone.

In other domains like natural language processing, it is well recognized that different users have unique tastes and requirements, and models can be adapted accordingly [38, 30]. Analogously, in image editing each user may exhibit distinct preferences - one user might favor images with brighter, more saturated colors, while another prefers muted tones and centered composition. Current text-to-image models and editors ignore these nuances, effectively optimizing for an average preference that may not align with any particular user. This lack of personalization means users must continuously correct or fine-tune outputs that do not suit their aesthetic, which limits the practicality of these AI editors in real creative workflows.

In this work, we take the first step towards *personalized image editing* by proposing a framework that learns and adapts to an individual user's editing preferences. Our method models how each user prefers their images to be modified, capturing aspects such as stylistic choices, color palettes, object attributes, and other consistently favored visual traits. By learning from user-specific preferences, our framework aligns image edits with the user's intent, producing results that better reflect their personal aesthetic. Importantly, we also recognize that collaboration across users can amplify personalization. Users often fall into clusters with overlapping tastes; by sharing preference information among like-minded users, we can generalize better from limited data. Consider a home-decor hobbyist who frequently edits interior photos to give them a rustic living-room feel. They always add a stone fireplace and distressed-leather sofa, but they have never thought to include exposed wooden ceiling beams. Several like-minded users in our preference graph routinely pair the fireplace-and-sofa combo with rough-hewn ceiling beams to complete the rustic look. Even though the target user has never requested the beams explicitly, our collaborative mechanism learns this object-level association from neighboring users' histories and can automatically insert the beams in future edits. The result is an image that remains true to the user's rustic touches while enriching the scene with an additional detail they are likely to appreciate.

In this paper, we propose a novel Direct Preference Optimization (DPO) framework for collaborative personalized editing. Specifically, we introduce a DPO-based learning approach that aligns a diffusion model's image edits with individual user preferences while simultaneously leveraging feedback from a community of users. To enable this capability, we propose a novel synthetic dataset of user editing preferences; a rich corpus of editing examples with annotations linking them to specific users and their satisfaction. We represent the users in this dataset as nodes in a graph, which captures similarities in their editing behavior. This graph-based collaborative learning, together with the DPO objective, allows our model to learn a wide spectrum of personalized editing styles within a single unified model. In summary, our contributions are three-fold:

- To the best of our knowledge, we introduce the first formulation of personalized text-to-image editing. We define a new problem setting where an editing model is tailored to an individual user's preferences, moving beyond the one-size-fits-all paradigm in image editing.

- We propose a novel training framework called, Collaborative Direct Preference Optimization, introducing a graph-structured regularization term into the DPO loss, explicitly modeling and leveraging collaborative relationships among user embeddings. This structured collaboration allows the model to capture nuanced preferences implicitly shared among like-minded users.

- We curate a novel synthetic dataset comprising 144K editing preferences which includes annotated examples of edits grouped by user. This dataset provides a valuable benchmark for studying personalization in image editing and facilitates quantitative evaluation of models in this setting.

- Our source code and dataset are publicly available at `http://personalized-editing.github.io`.

## 2 Related Work

**Text-to-Image Editing** Text-to-image diffusion models [36, 14, 34, 32] are frequently used for both image generation and editing because of their high-fidelity generation capabilities. Instruct Pix2Pix [5] enables flexibility across edit types with user prompts without requiring task-specific supervision, however, its performance can degrade for highly detailed or spatially complex edits. Blended Latent Diffusion [3] and SDEdit [23] provide more fine-grained control with diffusion models whereas ControlNet [42] conditions the model on additional structural inputs to maintain the spatial consistency while also allowing for localized edits.

**User Preference Optimization** Reinforcement Learning from Human Feedback (RLHF) [26] has been used to optimize Large Language Models (LLMs) [6, 29, 37] and multimodal LLMs [2, 22, 8, 1] to better align with human preferences. It involves training a reward model based on pairs of preferred and rejected data to guide the fine-tuning of the language model. A simpler alternative to RLHF is Direct Preference Optimization (DPO) [30], which directly optimizes the model parameters on preference data rather than the need for a separate reward model or reinforcement learning loop. The applications of DPO have extended beyond LLMs to vision domains. Adaptations such as Diffusion-DPO [38], have been proposed to similarly align diffusion models to generate images in accordance with human preferences. DPO has been used to further align models with individual users rather than an entire population [20].

**Personalized Image Generation** The expressive capabilities of T2I models have spurred research in user personalization. Recent methods have aimed to fit the preferences of an individual user. ViPER [35] fine-tunes a VLM on a dataset of AI-generated user comments on images to extract information on user preferences. It then includes these preferences in the prompt and classifier-free-guidance scale to align the model with the user during generation. However, the dependence on detailed comments provided by the user limits its ability to tailor specific image features in the absence of high-quality comments. Other methods have employed reward optimization to learn to represent user preferences in generation. PASTA [24] employs a reinforcement learning agent to iteratively refine a text prompt based on sequential user interactions before feeding to a T2I model. PPD [7] uses a VLM to compare liked and disliked images to train additional cross-attention layers using a Diffusion-DPO reward objective. Pigeon [40] trains a mask generator for user history and feeds to a multimodal LLM trained with a DPO objective to generate visual tokens used to guide the diffusion model. Although these methods aim to learn user preferences across individual history, they fail to consider that some user preferences may not exist within a user's own history, but can be inferred from that of similar or relevant users. Furthermore, none of these methods have addressed the task of user-personalized image *editing*, a much more complex task that requires maintaining consistency across edits while also capturing the nuanced editing preferences of users.

## 3 Background

The alignment of large language and diffusion models is often framed as a problem of *preference learning* in which a model is encouraged to rank preferred outputs higher than rejected ones. Traditional pipelines first train a separate *reward model* from pairwise human-preference data and then maximize the expected reward with reinforcement learning (RLHF) [26]. Although effective, RLHF adds algorithmic complexity, instability, and significant compute overhead.

**Direct Preference Optimization.** [30] recently introduced **Direct Preference Optimization** (DPO), a lightweight alternative that *removes* the explicit reward model and RL stage. Let $\mathcal{D} = \{(x_i, \ y_i^+, \ y_i^-)\}_{i=1}^N$ be a dataset of prompts $x_i$ paired with a **chosen** response $y_i^+$ and a **rejected** response $y_i^-$. Denote by $\pi_\theta(\cdot \mid x)$ the conditional distribution of the *trainable* policy and by $\pi_{\text{ref}}(\cdot \mid x)$ a frozen *reference* policy (e.g. the pre-SFT checkpoint). DPO maximizes the log–odds that the trainable policy assigns higher probability to $y^+$ than to $y^-$, while staying close to the reference. The resulting objective is a simple logistic loss

$$\mathcal{L}_{\text{DPO}}(\theta) \ = \ - \ \mathbb{E}_{(x,y^+,y^-)\sim\mathcal{D}} \Big[\log \sigma\big(\beta \, [\, \Delta_\theta(x, y^+, y^-) \ - \ \Delta_{\text{ref}}(x, y^+, y^-)]\big)\Big], \tag{1}$$

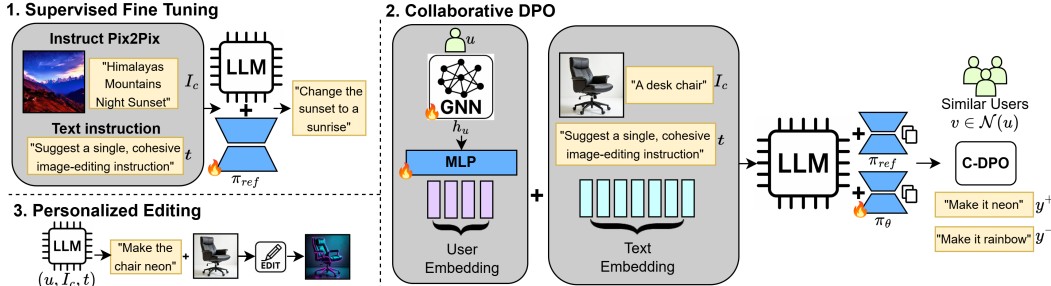

Figure 2: 1) We first fine-tune a language model so it can generate precise editing instructions. 2) We then introduce a graph-aware DPO objective that leverages collaborative user data to learn individual editing preferences. 3) After training, the system takes an input image and a user profile, produces tailored editing instructions, and outputs the corresponding personalized edit.

where $\sigma(\cdot)$ is the sigmoid, $\beta > 0$ is a temperature controlling conservatism, and $\Delta_\theta$ and $\Delta_{\text{ref}}$ are defined as

$$\Delta_\theta(x, y^+, y^-) = \log \pi_\theta(y^+ \mid x) - \log \pi_\theta(y^- \mid x), \tag{2}$$

$$\Delta_{\text{ref}}(x, y^+, y^-) = \log \pi_{\text{ref}}(y^+ \mid x) - \log \pi_{\text{ref}}(y^- \mid x). \tag{3}$$

**Text-to-image diffusion models** have become the leading framework for high-fidelity image generation from natural language inputs. These models iteratively denoise random noise into coherent images conditioned on text prompts, leveraging powerful encoders such as CLIP [28]. Foundational works like *DALL·E 2* [31], *Imagen* [34], and *Stable Diffusion* [32] have shown remarkable capabilities in producing semantically aligned and visually diverse images. These systems benefit from training on large-scale datasets using latent diffusion architectures that balance quality and efficiency. Extensions such as *InstructPix2Pix* [5] and *SDEdit* [23] introduce mechanisms for editing existing images through text-based instructions. However, these approaches remain general-purpose and do not adapt to individual users' stylistic preferences, limiting their applicability in personalized editing scenarios.

# 4 Methodology

We introduce Collaborative Direct Preference Optimization (C-DPO), a novel training framework that extends DPO by incorporating user-specific embeddings and graph-based collaboration for personalized image editing. Section 4.1 details our collaborative loss, which aligns edits with both individual and neighbor preferences, along with our user conditioning and training strategy. Section 4.2 describes personalized image editing via our diffusion-based backend. Fig. 2 gives an overview of our methodology.

## 4.1 Collaborative Direct Preference Optimization (C-DPO)

Direct Preference Optimization (DPO) has recently emerged as a stable and efficient alternative to RLHF for aligning models with human preferences. However, in its standard form, DPO is fundamentally limited for the task of personalized image editing. First, it lacks cross-user collaboration: all preference pairs are treated as if originating from a single anonymous user, leading the model to converge toward an average policy that ignores the diverse and often overlapping aesthetic preferences of real users. Second, as shown in Eq. 1, DPO assumes unimodal textual conditioning, which restricts its ability to incorporate structured, non-textual information such as user relationships or historical preferences encoded in a knowledge graph. These shortcomings prevent standard DPO from leveraging either personalization or collaboration, both of which are essential for effective user-specific image editing.

### 4.1.1 Modeling User-Edit Preferences in a Graph

To overcome the limitations of vanilla DPO, we extend the editing policy to explicitly condition on user identity and incorporate collaborative structure through a graph-based model of user preferences. We define the personalized editing policy as:

$$\pi_\theta(y \mid u, I_c, t), \tag{4}$$

where $u \in \mathcal{U}$ denotes the user identifier, $I_c$ is the caption describing the base image, and $t$ is the high-level editing instruction. Each user is represented by a learnable embedding $g_\phi(u) \in \mathbb{R}^d$, which encodes the user's stylistic preferences and conditions the model to generate personalized outputs.

To enable collaboration across users with overlapping preferences, we construct a heterogeneous, bipartite, undirected user–preference graph:

$$\mathcal{G} = (\mathcal{U}, \mathcal{P}, \mathcal{E}), \tag{5}$$

where $\mathcal{U}$ is the set of users initialized with a learnable embedding vector, $\mathcal{P}$ denotes preference attributes (e.g., color palettes, textures) initialized with a dense text embedding derived from a pretrained language model, and $\mathcal{E}$ consists of edges between users and their preferred or rejected attributes. This structure captures shared editing behavior and supports collaborative inference through shared interactions.

We encode the graph using a lightweight graph neural network (GNN), which aggregates neighbor information to compute a contextualized embedding $h_u$ for each user [12, 17]. Our GNN consists of a 2-layer GraphSAGE encoder and a decoder composed of 2 linear layers. We pretrain the GNN on an auxiliary supervised edge classification task by predicting user–attribute links as positive or negative, encouraging the encoder to learn semantically meaningful and stylistically distinct user embeddings.

A core benefit of our approach is that it generalizes to new users without requiring retraining, enabling both scalability and practicality. GraphSAGE inductively learns a set of aggregation functions that takes as input a node's neighborhood. When an unseen user arrives at inference, they are first added as a node with a zero-initialized feature vector in the existing graph and connected via edges to liked and disliked attributes. This new user's neighborhood is passed through the learned aggregation functions to compute the user's node embedding. While periodic fine-tuning as the graph expands may further refine embedding quality, it is not necessary for inference. Further GNN details can be found in Appendix B.

To quantify similarity between users, we compute a normalized similarity score between users $u$ and $v$ based on the number of shared one-hop neighbors (i.e., common preference attributes):

$$w_{uv} = \frac{|P(u) \cap P(v)|}{\max\limits_{u', v'} |P(u') \cap P(v')|}, \tag{6}$$

where $P(u) \subseteq \mathcal{P}$ is the set of attributes associated with user $u$. Here, $\mathcal{P}(u)$ denotes the set of preference attributes connected to user $u$, and the denominator is the maximum number of shared neighbors across all user pairs in the graph. These scores are later used as edge weights in our collaborative preference objective, allowing the model to softly attend to information from stylistically similar users. By conditioning the editing policy on graph-informed embeddings and user identity, our framework supports both fine-grained personalization and generalization through collaborative structure, addressing the core deficiencies of traditional DPO for personalized image editing.

### 4.1.2 Collaborative DPO Loss

To align the editing policy with both individual user preferences and insights from like-minded users, we introduce a collaborative extension to the DPO objective. The core idea is to preserve strong per-user alignment while softly regularizing the model using preference signals from a user's graph neighbors.

For a mini-batch of preference tuples, $(u, I_c, t, y^+, y^-)$ we define the Collaborative Direct Preference Optimization (C-DPO) loss as:

$$\mathcal{L}_{\text{C-DPO}} = \underbrace{\mathcal{L}_{\text{DPO}}(u, I_c, t, y^+, y^-)}_{\text{individual}} + \frac{\lambda}{\sum_{v \in \mathcal{N}(u)} w_{uv}} \sum_{v \in \mathcal{N}(u)} w_{uv} \underbrace{\mathcal{L}_{\text{DPO}}(v, I_c, t, y^+, y^-)}_{\text{collaborative}} \qquad (7)$$

where $\mathcal{N}(u)$ denotes the $K$ nearest neighbors of $u$ in $\mathcal{G}$, and $\lambda$ governs the strength of collaboration. Weighted averaging ensures that the collaborative term does not overpower the individual objective. Each $\mathcal{L}_{\text{DPO}}$ is instantiated via 1, but with the policy now conditioned on the *user information*. Because the reference policy $\pi_{\text{ref}}$ remains *user-agnostic*, the collaborative term pulls the user-conditioned policy toward neighbor preferences *only where the data supports it*, avoiding degenerate collapse to the global average. Equation 7 generalizes DPO to a *multi-user* setting. The first term preserves the per-user alignment benefits of vanilla DPO, whereas the second term implements a graph-regularized collaborative filter that shares statistical strength among like-minded users—crucial for data-sparse personalization.

### 4.1.3 Model Optimization Strategy

We adopt a two-stage training pipeline. First, we perform supervised fine-tuning (SFT) on paired base image and editing instruction data [5] to teach a pretrained language model to generate high-quality editing instructions conditioned on the image caption $I_c$ and text instruction $t$, but without any user information. Specifically, we train a LoRA adapter [15] on this task, enabling efficient adaptation of the base LLM for instruction generation. The resulting adapter is frozen and serves as the reference model $\pi_{\text{ref}}$ during subsequent preference-based training.

Afterwards, we fine-tune a separate copy of the same LoRA adapter using our collaborative DPO objective (Equation 7) over our user preference dataset. This adapter serves as the policy model $\pi_\theta$.

To personalize the model, we inject user-specific information as soft prompt tokens. For each user $u$, we compute a graph-based embedding $h_u$ using our GNN encoder. These embeddings are passed through a two-layer MLP and converted into a fixed set of soft tokens, which are prepended to the textual instruction $t$. This approach enables user conditioning without modifying the base language model architecture. Both the GNN and MLP components are trainable during C-DPO along with the LoRA, though we apply a lower learning rate to the GNN to preserve the structure learned during pretraining. During C-DPO training, the model learns not only from each user's preferences but also from the preferences of graph-connected neighbors, promoting better generalization across similar users while preserving individual alignment.

## 4.2 Image Editing with Personalized Editing Prompt

Once a personalized editing prompt is generated by our C-DPO model, we translate it into a concrete pixel-level transformation using a modular diffusion-based editing backend. We utilize FLUX [18], a large-scale rectified flow transformer developed for high-fidelity text-to-image generation, augmented with ControlNet [42] to ensure faithful, spatially constrained edits.

For a target user $u$, input image caption $I_c$, and high-level textual cue $t$, our tuned C-DPO model outputs a *personalized prompt*:
$$p_u = f_\theta(u, I_c, t),$$
where $f_\theta$ denotes the tuned C-DPO policy model. The output $p_u$ is a natural-language instruction tailored to user $u$'s preferences, and is directly consumable by any T2I-style sampler. For example: ``change the sofa's color to hot pink.''. This setup enables faithful and personalized image edits that reflect the user's intent with high semantic precision.

## 5 Experiments

We evaluate our personalized-editing framework with extensive qualitative and quantitative experiments. Qualitative experiments showcase how edits are personalized based on different users, while quantitative experiments measure how well the edited images reflect each user's unique style and preferences across thousands of (image, user) pairs.

**Experimental Setup** We evaluate our method's ability to generate personalized edits for a given base image. To model user-specific edit instructions, we fine-tune QWEN2.5 [27] using LoRA [15]. For image editing, we employ FLUX.1-DEV[2], enhanced with ControlNet[3] using a conditioning scale of 0.4 and Canny as the condition. We set the collaborative term strength, $\lambda$, to 0.15. While our full pipeline can be trained on a single NVIDIA L40 (48GB) GPU, we use multiple GPUs to accelerate training. During the initial supervised fine-tuning phase, we train for one epoch on the InstructPix2Pix training set [5] using 2 GPUs, completing in just under 3 hours. For collaborative tuning via C-DPO, we train for 3000 steps on our user editing preference dataset using 3 GPUs, with total training time of approximately 1 hour. We divide our dataset into training and test splits, with approximately 2,900 users in the training set and 100 users reserved for testing. Once trained, our system performs a personalized image edit for a given user in around one minute. Because our framework is editing-model agnostic, this process can be accelerated to about one second when paired with a faster backend such as TurboEdit [10]. Full training details and parameter configurations are provided in Appendix G.

**Dataset** To the best of our knowledge, no existing dataset captures the user-specific preferences necessary for personalized image editing. Therefore, we present a structured benchmark of 3,000 synthetic user profiles for studying personalized image editing in T2I diffusion models, spanning 144K samples on individual like/dislike preferences. To closely mimic the diversity of behavior, preferences and aesthetic tastes in the real world, we synthetically generate our data with distinct user demographic configurations that cover axes such as age, geography, and socioeconomic status. These configurations serve as a guide to the generation of detailed editing preferences, including themes, tones, overlays, likes, dislikes, and persona-aligned prompt examples. To simulate realistic editing interactions, each user is assigned four randomly sampled image captions from 80 COCO [21] categories. For each base image, we generate two pairs of preferred and rejected editing instructions across six edit types based on individual user preferences, yielding 48 annotated editing instructions per user. This rich data set serves as the foundation for training and evaluating our collaborative personalization framework. Our data set enables a rigorous evaluation of user alignment and preference modeling in diffusion-based editors. Our dataset is publicly available at `http://personalized-editing.github.io`.

## 5.1 Qualitative Experiments

We first showcase how identical objects are edited in distinct ways according to individual user preferences. Fig. 1, Fig. 3 (a) and Fig. 4 show several objects edited for different users with diverse profiles. Our system aligns each edit based on the likes and dislikes that the user has previously expressed. For instance, for a user (*labeled as 'Imaginative Child'*) who favors unicorns, rainbows, and vibrant, playful palettes (see Fig. 3 (a) and Fig. 4), our system infuses such preferences into a wide range of objects, from helmets and guitars to watchtowers. Our framework is also flexible to allow users to provide additional guidance while performing personalized edits. Fig. 3 (b) illustrates three distinct additional editing instructions such as *ice, lava, and monster* themes. As shown, our framework effectively combines user-specific preferences with the supplied guidance, producing coherent edits that align with both the explicit instruction and the underlying aesthetic of each user.

## 5.2 Quantitative Experiments

Since no prior work directly tackles personalized image editing, we evaluate our proposed framework along two complementary dimensions: a) Prompt-level alignment, measuring how well the editing instruction aligns with each user's liked and disliked concepts, b) Image-level alignment, assessing how closely the resulting edited image reflects the user's preferences based on user's liked and disliked concepts.

**Prompt-level alignment** We compare our method on a diverse set of strong baselines that vary in personalization capability, modeling assumptions, and training objectives. This comparison allows us to isolate the contributions of user conditioning and collaborative learning.

First, we provide two generic vision-language baselines: Qwen [27], a vanilla large-scale language model that takes only the image description as input and generates a generic edit prompt without awareness of user identity; and LLaVA [22], another open-source vision-language model that simi-

---

[2]`https://huggingface.co/black-forest-labs/FLUX.1-dev`
[3]`https://huggingface.co/InstantX/FLUX.1-dev-Controlnet-Union`

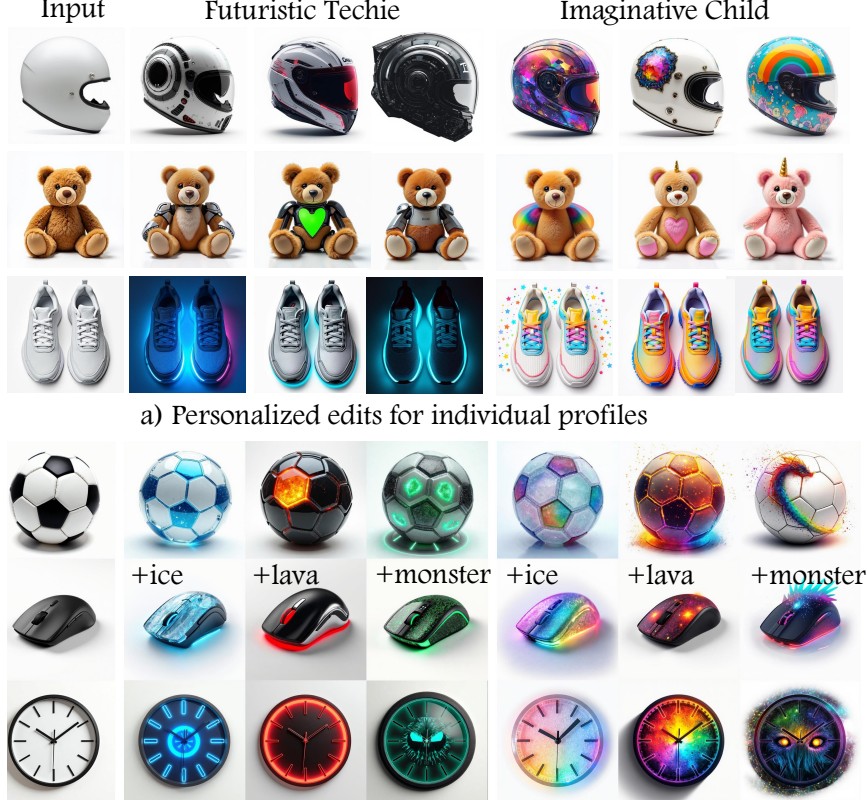

| Input | Futuristic Techie | Imaginative Child |

a) Personalized edits for individual profiles

+ice     +lava     +monster     +ice     +lava     +monster

b) Personalized edits with user-provided instructions

Figure 3: **(a) Qualitative Results for Individual Users on Different Objects.** Our framework is able to incorporate personalized elements into the image editing process, such as adding neon or futuristic elements for *Futuristic Techie* profile. **(b) Qualitative Results for User-Provided Personalized Edits.** Our framework allows users to provide additional guidance while performing personalized edits.

larly lacks any personalization signal and serves as a second reference point for non-user-specific performance. Both models act as task-agnostic baselines with no preference alignment.

Next, we consider Qwen (SFT), a fine-tuned variant trained on 150,000 generic edit-instruction pairs from the InstructPix2Pix dataset [5]. While this model is editing-aware, it lacks user conditioning, allowing us to assess the effect of task-specific fine-tuning alone. We also compare with Vanilla DPO [30], trained on pairwise preference data sampled from our dataset but without access to user identifiers or relational information. This model tests whether learning from global preference signals alone provides any personalization benefit. To evaluate the impact of incorporating explicit user identity, we also compare with a user-aware variant, DPO-User, that uses a learnable embedding for each user ID but does not leverage any collaborative structure. This setup mimics the PPD objective [7] and represents a personalization mechanism based on private embeddings alone. Finally, our proposed method - Collaborative-DPO extends DPO-User by introducing graph-based collaboration, allowing user representations to evolve through interactions with stylistically similar neighbors. This model constitutes the full version of our framework.

To compare these methods fairly, we standardize the prompting format across all models. Each method receives an input of the form: "Given this user's profile <profile>, suggest an edit for the following image: <image description>." We construct three evaluation conditions reflecting different profile completeness: (i) Like+Dislike, where both liked and disliked examples are available; (ii) Likes-only; and (iii) Dislikes-only. For each condition, we measure CLIP-text similarity between the generated instruction and the user's ground-truth liked instructions. As shown in Table 2, our C-DPO

|     | Original | Edit 1 | Edit 2 | Edit 3 | Edit 4 |
|-----|----------|--------|--------|--------|--------|

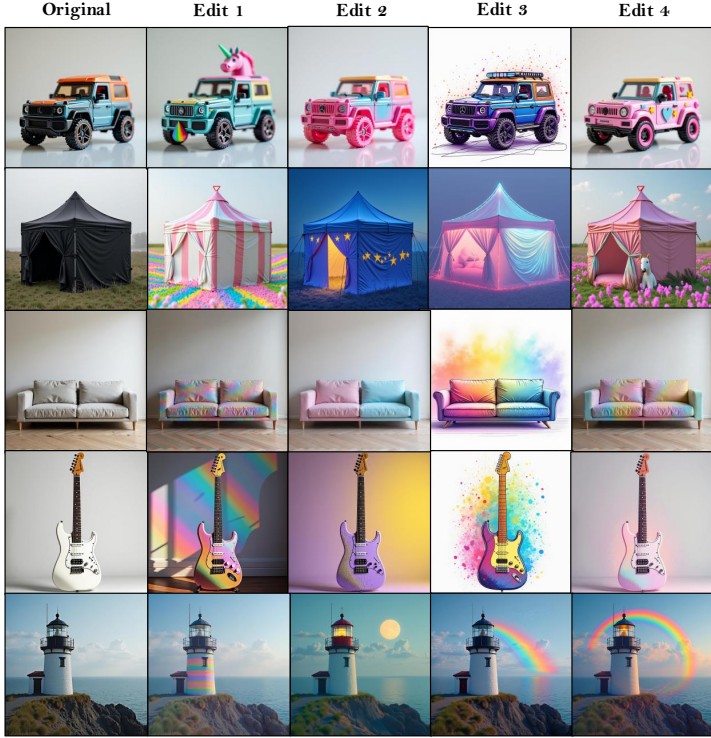

Figure 4: **Qualitative Results for a Single User on Diverse Objects.** For a user who loves unicorns, rainbows, and vibrant, playful palettes, our system infuses that whimsical aesthetic into a wide range of objects - from cars and guitars to watchtowers.

model achieves the highest score across all configurations, indicating its ability to recover user intent from partial or complete preference signals.

**Image-level alignment** Next, we report DINO and CLIP scores computed between the original input image and its personalized edited version (see Table 3). The results indicate that our method better preserves the original identity compared to other approaches. We also compute HPS and CLIP-T scores between the personalized edited images and the textual description of the original image, further demonstrating that our method achieves higher alignment with the original content than competing methods. Please refer to Appendix A for additional quantitative experiments and comparisons with state-of-the-art editing methods.

**Ablations** Please refer to Appendix C for ablations on collaborative scale parameter $\lambda$, neighborhood size $K$, choice of editing method, and prompt-engineering baseline.

**Personalized Image Generation** Our method can also be repurposed for personalized image *generation* as opposed to *editing*. Please visit A.2 for more details.

### 5.3 User Study

We further validate our results through a user study to assess perceptual personalization quality. We ran a crowd-sourced evaluation on Prolific.com with 50 participants. We randomly sampled 10 synthetic user profiles from our test split to cover a range of stylistic tastes. For each profile we first showed a short text description of what aspects this user likes and dislikes based on their previous edit history. Participants then judged three target images that had been edited

Table 1: User Study Results

| Model | Q1 | Q2 |
|-------|------|------|
| DPO-Vanilla | 0.210 | 0.210 |
| DPO-User | 0.174 | 0.325 |
| Ours | **0.616** | **0.465** |

by three competing methods (Vanilla DPO, DPO-User and Ours). Each edited image was displayed alongside its text prompt; the order of methods and the order of questions were fully randomized per task to prevent position bias.

Table 2: CLIP scores between model outputs and user preferences under different prompting conditions where we provided both Like and Dislike preferences of the user (Like+Dislike), only like preferences (Likes) or only dislike preferences (Dislikes).

| Model | Like+Dislike | Likes | Dislikes |
|---|---|---|---|
| Qwen-VL | $0.274 \pm 0.055$ | $0.276 \pm 0.052$ | $0.229 \pm 0.051$ |
| LLaVA | $0.244 \pm 0.065$ | $0.254 \pm 0.061$ | $0.227 \pm 0.051$ |
| SFT-Vanilla | $0.276 \pm 0.049$ | $0.276 \pm 0.042$ | $0.185 \pm 0.065$ |
| SFT-Pix2Pix | $0.227 \pm 0.078$ | $0.238 \pm 0.065$ | $0.196 \pm 0.065$ |
| DPO-Vanilla | $0.272 \pm 0.059$ | $0.274 \pm 0.057$ | $0.172 \pm 0.057$ |
| DPO-User | $0.307 \pm 0.071$ | $0.286 \pm 0.071$ | $0.269 \pm 0.076$ |
| Ours | $\mathbf{0.354 \pm 0.068}$ | $\mathbf{0.306 \pm 0.069}$ | $\mathbf{0.294 \pm 0.071}$ |

Table 3: DINO and CLIP scores computed between the original input image and its personalized edited version (DINO-Ref, CLIP-Ref). HPS and CLIP-T scores are computed between the personalized edited images and the textual description of the original image.

| Model | DINO-Ref | CLIP-Ref | HPS | CLIP-T |
|---|---|---|---|---|
| SFT | $0.690 \pm 0.263$ | $0.600 \pm 0.222$ | $0.238 \pm 0.044$ | $0.331 \pm 0.075$ |
| DPO-Vanilla | $0.736 \pm 0.248$ | $0.615 \pm 0.202$ | $0.242 \pm 0.049$ | $0.346 \pm 0.067$ |
| DPO-User | $0.762 \pm 0.240$ | $0.649 \pm 0.224$ | $0.243 \pm 0.048$ | $0.354 \pm 0.066$ |
| **Ours** | $\mathbf{0.782 \pm 0.182}$ | $\mathbf{0.652 \pm 0.182}$ | $\mathbf{0.249 \pm 0.034}$ | $\mathbf{0.358 \pm 0.043}$ |

Adapted from the user study of [38], workers were asked the following questions for every image: Q1: General Preference – "Which edited image would this user prefer, given the prompt?" Q2: Visual Appeal – "Ignoring the prompt, which image is more visually appealing?". Table 1 summarizes responses from the 50 participants across the two evaluation questions. In every category our method secures the highest win rate by a large margin. These results indicate that, human judges consistently favor the edits produced by our method. We further conduct a user study with real-user preference data, replacing the synthetic preferences used in earlier experiments. In this study, participants provided their own liked and disliked concepts, and we evaluated how well the edited images aligned with these individual preferences. Please refer to Appendix E for detailed methodology and results.

## 6 Discussion

**Broader Impact and Limitations** By aligning text-to-image diffusion models to each individual's aesthetic, our framework can lower the barrier to high-quality visual content creation, reducing repetitive prompt-engineering cycles and enabling non-experts, including artists with motor impairments or limited technical skills to achieve their desired edits more efficiently. On the negative side, since our system tailors outputs to inferred tastes, it also risks reinforcing aesthetic "filter bubbles," narrowing users' exposure to diverse visual styles. In terms of technical limitations, when a new user lacks both personal edits *and* close neighbors in the graph, our model defaults to a generic editing. Furthermore, because our framework depends on off-the-shelf models such as Flux and ControlNet, any biases embedded in those backbones may propagate to the resulting edits. As for data, we choose to generate our data synthetically because of the scalability and budget constraints of crowd-sourcing, however, obtaining real-world user preference data presents an exciting future research direction by gathering more authentic and nuanced information.

**Conclusion** We presented the first framework that unifies *personal* and *collaborative* preference learning for editing with text-to-image diffusion models. Our novel objective function C-DPO (i) models each user's preference through learned graph embeddings, (ii) propagates information across a similarity graph of like-minded users, and (iii) couples the resulting personalized prompt with a T2I diffusion model to deliver high-fidelity, user-aligned edits. Our study demonstrates that collaborative signals can be harnessed *without* sacrificing individual customization, paving the way for practical editing assistants that adapt fluidly to diverse aesthetics. Future research will focus on extending the framework beyond still images such as *video domain*, where temporal coherence and multi-object consistency introduce new challenges, which remains an exciting direction for follow-up work.

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

Table 4: We report pairwise similarity scores between the user's liked images and the generated personalized images (image–image similarity using DINO-Pref and CLIP-I-Pref), as well as between the personalized edited images and the user's liked prompts (image–text similarity using CLIP-T-Pref).

| Model | DINO-Pref | CLIP-I-Pref | CLIP-T-Pref | CLIP-T-Neighbor |
|---|---|---|---|---|
| SFT | $0.054 \pm 0.102$ | $0.315 \pm 0.098$ | $0.334 \pm 0.049$ | $0.286 \pm 0.025$ |
| DPO-Vanilla | $0.056 \pm 0.106$ | $0.335 \pm 0.097$ | $0.359 \pm 0.035$ | $0.304 \pm 0.026$ |
| DPO-User | $0.056 \pm 0.105$ | $0.325 \pm 0.096$ | $0.359 \pm 0.034$ | $0.297 \pm 0.026$ |
| **Ours** | $\mathbf{0.059 \pm 0.106}$ | $\mathbf{0.338 \pm 0.089}$ | $\mathbf{0.364 \pm 0.038}$ | $\mathbf{0.312 \pm 0.020}$ |

Table 5: Quantitative comparison for personalized image generation with VIPER.

| Model | CLIP-I | DINO | HPS |
|---|---|---|---|
| ViPer | $0.286 \pm 0.015$ | $0.035 \pm 0.008$ | $0.205 \pm 0.050$ |
| **Ours** | $\mathbf{0.326 \pm 0.020}$ | $\mathbf{0.037 \pm 0.016}$ | $\mathbf{0.253 \pm 0.046}$ |

# A    Additional Quantitative Experiments

We conduct additional quantitative experiments to evaluate our personalized image editing framework. Specifically, we compare user-preferred images with the personalized outputs generated by our method using DINO [25], CLIP-I, CLIP-T [28], and HPS [39] metrics (see Sec. A.1).

Furthermore, we demonstrate the effectiveness of our approach for personalized image generation by benchmarking against state-of-the-art methods in this domain (see Sec. A.2).

## A.1    Experiments on Personalized Image Editing

Table 2 (main paper) presents a quantitative comparison between our method and other state-of-the-art approaches using editing prompts generated by each method, as these baselines only support prompt-level generation rather than direct image outputs. To further evaluate image-level quality, we conduct additional experiments comparing SFT, Vanilla DPO, and User-based DPO against our method. For each method, we generated editing prompts and used Flux ControlNet to produce corresponding images. Results of the quantitative comparisons are shown in Table 4 and Table 3.

First, in Table 4, we report pairwise similarity scores between the user's liked images and the generated personalized images (image–image similarity using DINO-Pref and CLIP-I-Pref), as well as between the personalized edited images and the user's liked prompts (image–text similarity using CLIP-T-Pref). We also include a CLIP-T-Neighbor measure, which evaluates how well the personalized images align with attributes favored by similar users. For each user, we identify their ten most similar users based on the user-preference graph, aggregate the those users liked attributes, and compute the CLIP similarity between these aggregated preferences and the images generated for the target user. As can be seen from the results, our method achieves better results in generating personalized editing images.

## A.2    Experiments on Personalized Image Generation

To further validate the effectiveness of our method beyond personalized image editing, we compare it with the state-of-the-art personalized image generation method VIPER [35]. As shown in Table 9 and Fig. 9, our method achieves higher scores when comparing generated images to the user's liked images. Additionally, it delivers superior visual quality and relevance, better aligning with user preferences (see Fig. 9).

## A.3    Comparison with State-of-the-Art Instruction-Based Editing Models

To better contextualize our approach within the broader landscape of instruction-based and unified image-editing frameworks, we conducted additional quantitative comparisons against recent state-of-

Table 6: Quantitative comparison with recent state-of-the-art instruction-based editing frameworks. Our C-DPO method achieves superior personalization and overall alignment

| Method | CLIP-I-Pref | DINO-Pref | HPS |
|---|---|---|---|
| BAGEL | $0.3397 \pm 0.0144$ | $0.0363 \pm 0.0066$ | $0.2475 \pm 0.0388$ |
| SEED-Llama | $0.3109 \pm 0.0159$ | $0.0349 \pm 0.0121$ | $0.2500 \pm 0.0287$ |
| Nexus-Gen | $0.3327 \pm 0.0138$ | $\mathbf{0.0373 \pm 0.0113}$ | $0.2598 \pm 0.0307$ |
| Ours | $\mathbf{0.3515 \pm 0.0190}$ | $0.0365 \pm 0.0164$ | $\mathbf{0.2607 \pm 0.0328}$ |

the-art baselines: BAGEL [9], SEED-LLAMA [11], and Nexus-Gen [41]. For fairness, we provided each model with the same textual user-preference prompts used in our setup, thereby allowing them to leverage equivalent conditioning information without requiring architectural modification.

As summarized in the Table 9, our approach clearly outperformed these methods across standard image editing evaluation metrics, highlighting the distinct advantage of our collaborative personalization framework.

## B  Detailed GNN Information

Here, we go into more detail regarding graph architecture, GNN construction and training, and generalization to unseen users at inference.

### B.1  GNN Construction and Training

We employ a lightweight Graph Neural Network (GNN) to compute contextualized user embeddings by aggregating neighborhood information in a heterogeneous bipartite graph of users and attributes. Each attribute node represents an editing instruction topic (e.g., rustic themes, neon colors, minimalistic textures) and is initialized with a dense text embedding derived from a pretrained language model. Each user node is initialized with a learnable embedding vector. The resulting graph encodes user–attribute relationships via edges that indicate whether a user likes or dislikes a given attribute.

Our GNN architecture consists of a 2-layer GraphSAGE convolutional encoder using mean aggregation [12], followed by an edge decoder comprising two linear layers. Details on specific layer dimensions can be found in Table 12 The model is pretrained on an auxiliary edge-prediction task designed to classify whether a user–attribute link is positive or negative. This task encourages the GNN to learn semantically meaningful user representations grounded in the structure of the user–attribute graph. We train the model on 60% of the edges, reserving 20% each for validation and testing, following standard protocols outlined in foundational works [12, 17].

During pretraining, both liked and disliked user–attribute pairs are leveraged in a supervised edge classification task (e.g., edges of the form (user1, likes, "neon green") or (user2, dislikes, "rainbows")). Once pretraining is complete, the classification head is discarded and only the encoder's user embeddings are retained. These embeddings are subsequently used in the full Collaborative Direct Preference Optimization (C-DPO) stage, where user embeddings are combined with target image captions (e.g., "White ceramic bowl") to construct user-specific positive and negative instruction pairs (e.g., "Add a holographic pattern to the bowl" as positive, "Overlay a floral pattern on the bowl" as negative).

### B.2  Generalizing to Unseen Users

A central advantage of our approach is that it supports generalization to new users without retraining. Our approach follows an inductive embedding framework (GraphSAGE) rather than a transductive one. Unlike transductive methods, which are limited to fixed graphs, GraphSAGE learns a set of aggregation functions that generalize to unseen nodes by aggregating features from their neighborhoods. This allows our model to efficiently infer embeddings for new users.

At inference time, when a new users arrives, (1) we create a new user node with a zero-initialized feature vector of fixed width, (2) we connect this node to relevant attribute nodes representing the user's liked and disliked attributes. (3) We apply the pretrained GNN encoder's learned aggregation

Table 7: Ablation on C-DPO Neighborhood Size ($K$)

| $K$ | CLIP Liked Similarity |
|---|---|
| 2 | $0.353 \pm 0.065$ |
| 3 | $0.354 \pm 0.068$ |
| 5 | $0.358 \pm 0.065$ |
| 12 | $0.344 \pm 0.069$ |

functions over this new node's neighborhood to compute the user's embedding inductively from the connected attributes and their neighborhood structure.

This process requires no retraining, making it scalable and practical for continual deployment. Although periodic fine-tuning on the expanded graph may further refine embedding quality, it remains optional and is not necessary for inference.

# C Additional Ablations

## C.1 Ablation on Collaborative Scale Parameter

We conduct an ablation study on the collaborative scaling factor $\lambda$, evaluating values of $0.01$, $0.15$, and $0.50$ (see Fig. 7). A very low value ($\lambda = 0.01$) limits the influence of neighbor preferences, effectively reducing the method to a user-based DPO, as neighbors contribute insufficient information to guide personalization. In contrast, a high value ($\lambda = 0.50$) overly emphasizes neighbor preferences, which can override the user's own preferences and degrade performance. Our results indicate that $\lambda = 0.15$ provides the best balance, allowing meaningful influence from similar users without overwhelming the target user's preferences.

## C.2 Ablation on Neighborhood Size

We further investigate the effect of the collaborative neighborhood size $K$, which determines how many of the most similar users contribute weighted preference signals in the C-DPO loss 7. As shown in Table 7, smaller to moderate values yield stronger alignment between generated edits and user preferences, as measured by CLIP Like Similarity. Increasing $K$ beyond a certain threshold leads to diminishing returns or even degraded performance, likely due to noise from less relevant neighbors. Although the synthetic nature of our dataset may influence this, the phenomenon is not exclusive to synthetic data and generally occurs in collaborative and graph-based methods, even with real-world datasets [13, 43, 19]. When too many neighbors are aggregated, weaker or less relevant signals tend to dilute the informative collaborative signals from closely related neighbors—a well-documented issue across various real-world collaborative filtering and recommendation tasks.

## C.3 Ablation on Editing Method

In our experiments, we used Flux ControlNet as the base editing method. However, our method is applicable to Stable Diffusion-based editing methods as well. To highlight the impact of the underlying editing method, we compared personalized editing results across several Stable Diffusion-based approaches: SDEdit [23], Ledits++ [4], InstructPix2Pix [5], TurboEdit [10] as well as Flux-based methods including RF Inversion [33] and Flux ControlNet[4]. As shown in Fig. 8, Flux ControlNet consistently delivers superior fidelity and visual quality compared to the other Flux-based methods. Similarly, our personalized editing with Ledits++ performs better than other SD based methods.

## C.4 Ablation on Prompt-Engineering Baseline

To further justify the added complexity of our GNN-based collaborative personalization framework, we compare C-DPO against a strong prompt-engineering baseline that directly injects textual user descriptions, including aggregated neighbor information, into the model prompt, without using

---

[4]`https://huggingface.co/InstantX/FLUX.1-dev-Controlnet-Union`

Table 8: Comparison between our C-DPO framework and a prompt-engineering baseline that encodes user and neighbor information textually. Our method achieves markedly higher user-alignment scores, validating the effectiveness of GNN-based personalization.

| Method | Like+Dislike | Likes | Dislikes |
|---|---|---|---|
| Prompt Engineering | $0.270 \pm 0.051$ | $0.270 \pm 0.045$ | $0.200 \pm 0.057$ |
| Ours | $\mathbf{0.354 \pm 0.068}$ | $\mathbf{0.306 \pm 0.069}$ | $\mathbf{0.294 \pm 0.071}$ |

Table 9: Quantitative comparison with recent state-of-the-art instruction-based editing frameworks. Our C-DPO method achieves superior personalization and overall alignment

| Method | CLIP-I-Pref | DINO-Pref | HPS |
|---|---|---|---|
| BAGEL | $0.3397 \pm 0.0144$ | $0.0363 \pm 0.0066$ | $0.2475 \pm 0.0388$ |
| SEED-Llama | $0.3109 \pm 0.0159$ | $0.0349 \pm 0.0121$ | $0.2500 \pm 0.0287$ |
| Nexus-Gen | $0.3327 \pm 0.0138$ | $\mathbf{0.0373 \pm 0.0113}$ | $0.2598 \pm 0.0307$ |
| Ours | $\mathbf{0.3515 \pm 0.0190}$ | $0.0365 \pm 0.0164$ | $\mathbf{0.2607 \pm 0.0328}$ |

GNN-derived embeddings. This baseline represents a simplified but potentially competitive approach where user and neighborhood preferences are expressed through structured textual conditioning rather than learned representations.

As shown in Table 8, our proposed method substantially outperforms this baseline across all metrics. These results indicate that our proposed framework provides a meaningful benefit over simpler, prompt-based personalization methods, effectively justifying the additional complexity introduced by the GNN.

# D    Additional Qualitative Experiments

## D.1    Additional Qualitative Examples

We first showcase how identical objects are edited in distinct ways according to individual user preferences. Fig. 5 shows several objects edited for different users with diverse profiles. Our system adapts each image to the likes and dislikes that the user has previously expressed (see 'User Profile' which describes user's preferences). For instance, the first row depicts a user who prefers playful elements, vivid color palettes and childish elements such as unicorns and rainbows. When this user is presented with a bike or a ceramic bowl, our model automatically adds rainbow-hued paint, pastel gradients, and subtle unicorn-themed playful touches that align with a child-friendly aesthetic, while preserving the underlying object structure in a disentangled manner. On the other hand, when our system is presented with a user who prefers earthy tones and nature, our framework adds natural elements such as leaves, wooden materials and plants.

## D.2    Visuals on Personalized Image Generation

We further demonstrate the applicability of our method for personalized image generation by comparing it with the state-of-the-art approach, VIPER [35]. Fig. 9 presents results for two distinct user profiles: the Imaginative Child, who prefers rainbow colors and glitter, and the Glamour Stylist, who favors shiny and metallic elements. As shown, our method captures a broader range of user-specific preferences while maintaining focus on coherent objects. In contrast, VIPER often emphasizes textures or patterns without anchoring them to meaningful objects.

## D.3    Comparison with State-of-the-Art Instruction-Based Editing Models

To better contextualize our approach within the broader landscape of instruction-based and unified image-editing frameworks, we conducted additional quantitative comparisons against recent state-of-the-art baselines: BAGEL [9], SEED-LLAMA [11], and Nexus-Gen [41]. For fairness, we provided each model with the same textual user-preference prompts used in our setup, thereby allowing them to leverage equivalent conditioning information without requiring architectural modification.

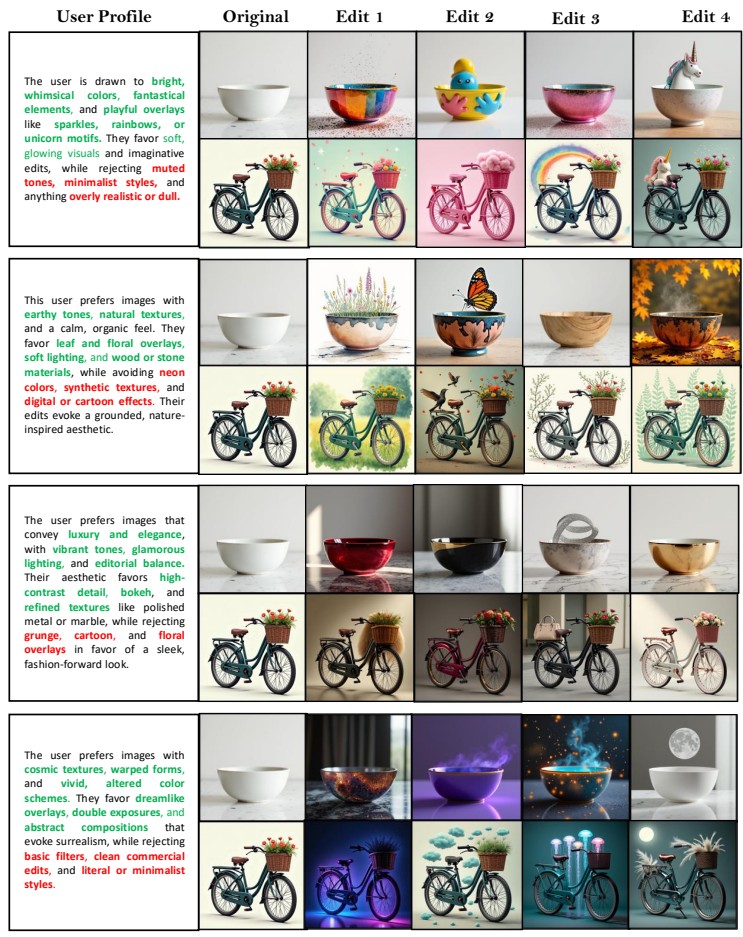

Figure 5: **Qualitative Results for Different Users on the Same Objects.** Our framework tailors image edits to each user's preferences. In the first row, for example, a user who loves unicorns, rainbows, and playful color schemes sees their inputs transformed to match that personalized aesthetic preferences.

As summarized in the Table 9, our approach clearly outperformed these methods across standard image editing evaluation metrics, highlighting the distinct advantage of our collaborative personalization framework.

# E    Additional User Studies

We conducted a two-stage user survey with 10 participants using real-human data. Specifically, we first asked participants to indicate their likes and dislikes attributes from a predefined list of 45 concepts as suggested. This list covers a wide range of colors (e.g. standard colors, neon colors), styles (e.g. watercolor, cyberpunk), textures (e.g. glossy, wood) and other concepts (such as rainbow, butterfly).

Based on these preferences, we generated personalized images using our proposed method. Participants were then sent a follow-up survey featuring personalized edits of five objects: an apple, a mug, a bag, a teddy bear, and a bowl. In this survey, participants viewed both the original and personalized versions of each object and answered the following questions:

Q1: On a scale of 1 to 5 (1=Does not matching my personal taste at all, 5=This is matching my personal taste a lot) can you rate the following image?

Q2: To what extent does the edit preserve the original image while reflecting personalized preferences? (1=Not at all, 5=Very Much)

Table 10: Additional User Study results.

| Object Name | Q1: Personalization | Q2: Disentanglement |
|---|---|---|
| Apple | $4.375 \pm 0.74$ | $4.75 \pm 0.46$ |
| Mug | $4.625 \pm 1.06$ | $4.50 \pm 0.75$ |
| Teddy bear | $4.75 \pm 0.46$ | $4.625 \pm 0.51$ |
| Bag | $4.50 \pm 1.06$ | $3.85 \pm 0.83$ |
| Bowl | $4.75 \pm 0.46$ | $5.00 \pm 0.00$ |
| Total | $\mathbf{4.60 \pm 0.15}$ | $\mathbf{4.55 \pm 0.38}$ |

Before Personalization        After Personalization

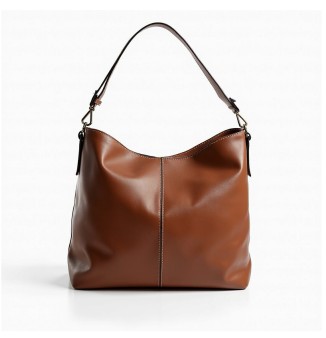 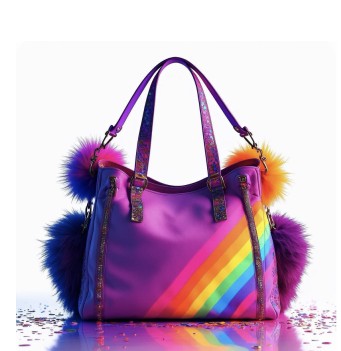

Figure 6: Example edit for a member of our real user study. *This user preferred pink, purple, and neon colors, while disliking white, gray, and beige. Their favored patterns included rainbow, fluffy, and glossy styles, while they disliked floral and earthy designs.* This image was rated as Personalization = 5/5, Disentanglement = 4/5, showing that the user found the edit matching their taste.

Our results (Table 10) demonstrate that real users found the images personalized by our framework highly personalized and disentangled.

On the same real-user preference data, we conducted an additional user study with personalized edits for each image to compare baselines. Specifically, users are presented with an object alongside three personalized edits (from Vanilla DPO, User-DPO, and our C-DPO method) where images are shown in an anonymous fashion. Users are asked to pick the edit they most prefer out of 3 options.

Table 11: Additional User Study comparison of preference ratios across models.

| Object | C-DPO (Ours) | User-DPO | User-Vanilla |
|---|---|---|---|
| A Bag | 0.45 | 0.22 | 0.33 |
| A Bowl | 1.00 | 0.00 | 0.00 |
| A Mug | 0.45 | 0.33 | 0.22 |
| A Teddy Bear | 0.78 | 0.22 | 0.00 |
| An Apple | 1.00 | 0.00 | 0.00 |

## F  Evaluation Prompt

To ensure a fair comparison across baselines, we adopt a unified structured prompt for all evaluations (see Fig. 10).

## G  Experimental Details

Various details of our experimental setup are given in Table 12.

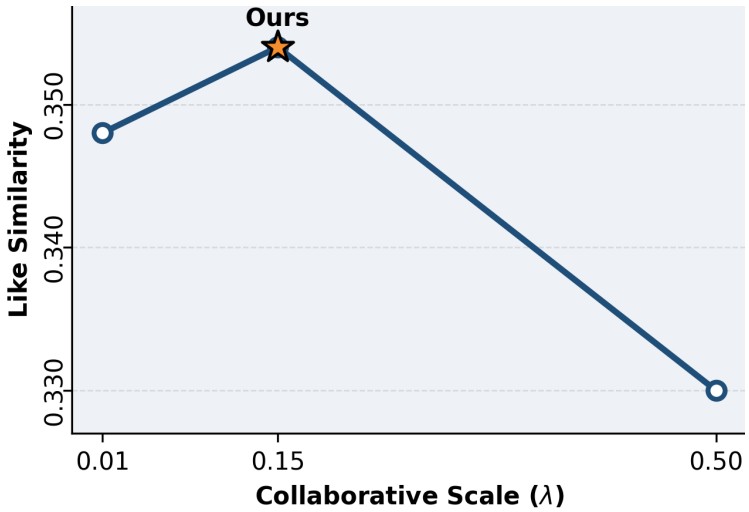

Figure 7: Ablation on Collaborative Scale ($\lambda$)

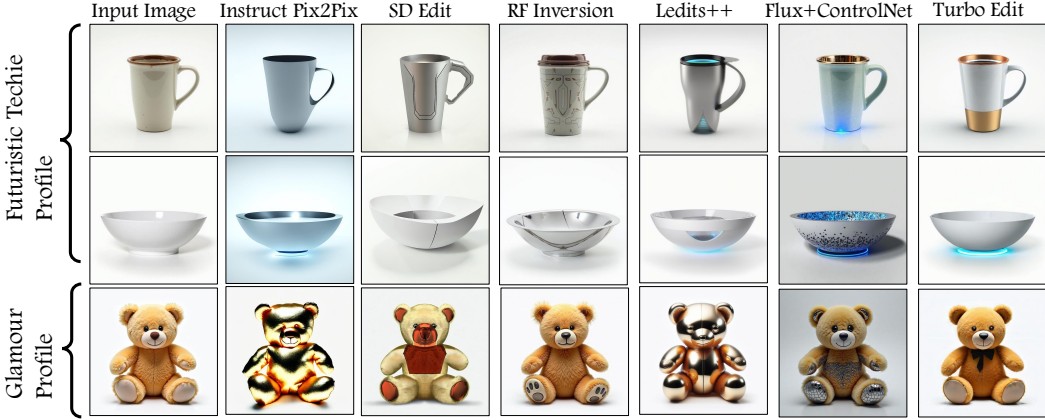

Figure 8: Ablation on Stable Diffusion and Flux based editing methods.

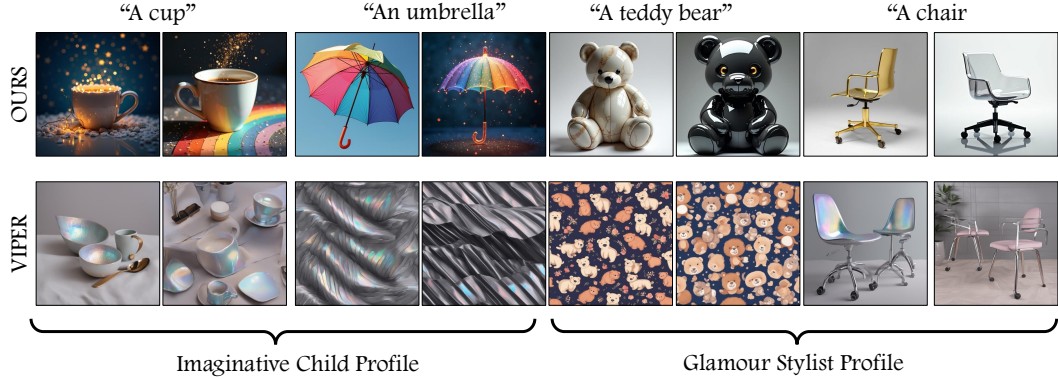

Figure 9: Personalized image generation for two profiles: Imaginative Child Profile who likes rainbow colors and glitter, and Glamour Stylist Profile who likes shiny and metallic elements.

*System Prompt*

You are an expert at providing user personalized image-editing instructions.

*User Prompt*

Likes: [e.g. colorful and playful styles, unicorn motifs, rainbows]
Dislikes: [e.g. minimalistic styles, muted tones, overly realistic imagery]

Original Image Caption: [e.g. "A white ceramic bowl"]

Suggest a single, cohesive image-editing instruction.

Figure 10: The input prompt template used during our evaluations.

**Which edited image would this user prefer, given their preferences?** *

**User Preferences:**

👍 **Likes:**

Futuristic | Sci-fi | Neon colors | Glass & metal textures | Holographic & circuit patterns | LED/glow effects | Cyberpunk/digital styles

👎 **Dislikes:**

Vintage | Sepia/pastels | Wood/fabric textures | Floral/polka patterns | Traditional art styles

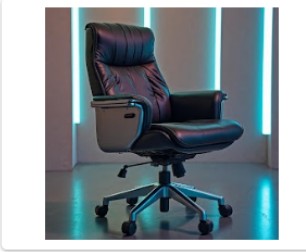

○ Option 2

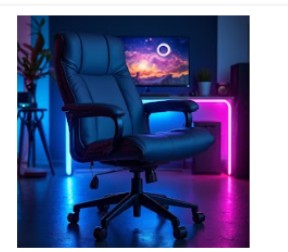

○ Option 3

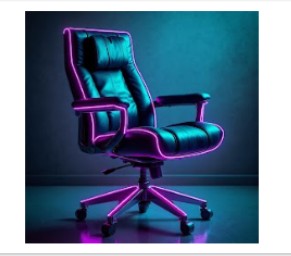

○ Option 1

Figure 11: User Study Q1

**Which image is more visually appealing?** *

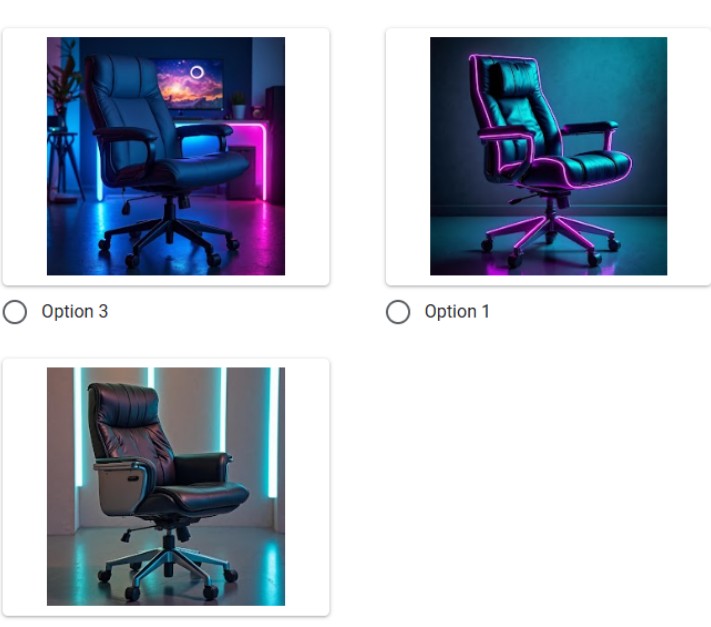

○ Option 3                    ○ Option 1

○ Option 2

Figure 12: User Study Q2

| LoRA Details | |
| --- | --- |
| Base LLM | Qwen/Qwen2.5-7B-Instruct [27] |
| LoRA Rank | 16 |
| LoRA $\alpha$ | 32 |
| Dropout | 0.05 |
| Optimizer | AdamW |
| LR Scheduler | Cosine |
| Data Type | bfloat16 |
| Target Modules | All Linear |

| User Based GNN | |
| --- | --- |
| Embedding Size | 4096 |
| Keyword Encoder | Linq-AI-Research/Linq-Embed-Mistral [16] |
| Pretrain Epochs | 150 |
| Pretrain Learning Rate | 3e-5 |
| Pretrain Optimizer | Adam |
| GraphSAGE Hidden Channels | 1024 |
| Linear Hidden Channels | 1024 |

| Initial Supervised Finetuning | |
| --- | --- |
| Data | Instruct Pix2Pix [5] |
| GPUs | 2 NVIDIA L40 (48GB) GPUs |
| Train Time | ∼3 Hours |
| Epochs | 1 |
| Learning Rate | 2e-6 |
| Batch Size | 16 |

| Collaborative DPO | |
| --- | --- |
| Data | Our Synthetic Data |
| GPUs | 3 NVIDIA L40 (48GB) GPUs |
| Start Checkpoint | SFT Tuned LoRA on Instruct Pix2Pix |
| Train Time | ∼1 Hour |
| Number of Similar Users | 3 |
| Collaborative Term Scale ($\lambda$) | 0.15 |
| Number of Soft Prompt Tokens | 8 |
| Train Steps | 3000 |
| Learning Rate | 2e-7 |
| GNN Parameters Learning Rate | 2e-8 |
| Batch Size | 12 |

| Flux+ControlNet | |
| --- | --- |
| Flux Model | black-forest-labs/FLUX.1-dev [18] |
| ControlNet | InstantX/FLUX.1-dev-Controlnet-Union |
| Control Mode | Canny |
| Number of Inference Steps | 60 |
| Guidance Scale | 35 |
| ControlNet Conditioning Scale | 0.3-0.4 |

Table 12: Full Experimental Details for Replicating Our Method

