# OpenReview forum: "Personalized Image Editing in Text-to-Image Diffusion Models via Collaborative Direct Preference Optimization"
_NeurIPS.cc/2025/Conference — NeurIPS 2025 poster_

### Official Review · Reviewer_HyiN · 2025-06-24

**Clarity:** 3
**Significance:** 2
**Originality:** 3
**Rating:** 3
**Confidence:** 4

**Summary:**

The paper introduces **Collaborative Direct Preference Optimization (C-DPO)**, a novel framework designed to address the fundamental limitation of generic text-to-image (T2I) diffusion models by enabling **personalized image editing**. C-DPO uniquely aligns image edits with **individual user preferences** by modeling users in a dynamic preference graph and **leveraging collaborative signals** from like-minded individuals with overlapping tastes. It integrates user-specific embeddings, learned via a lightweight graph neural network, into a **novel DPO objective** that jointly optimizes for both individual user alignment and neighborhood coherence. A key contribution is the **first formulation of personalized T2I editing**, supported by the creation of the **first large-scale synthetic benchmark** comprising 144K richly structured editing preferences.

**Questions:**

1. Novelty (same as weakness 2): The technique novelty referring to the C-DPO is low, which I would like the authors to claim more.

2. Qualitative Comparisons (as weakness 3): I don't know whether it's allowed to put any anonymous links to some image collections during rebuttal. However, quantitative comparison with some unified or image-editing models may help to claim your advantages over them on the personalized image editing task.

**Ethical Concerns:**

["NO or VERY MINOR ethics concerns only"]

**Final Justification:**

The core equation is Eq.7 in the paper, which is too simple as also widely used in federated learning several years ago. This paper may get better rating if the main context is enriched with more qualitative comparisons with image editing methods. Given its current shape, I cannot convince myself to raise the score. It's not very matching the NeurIPS bar.

**Limitations:**

Please check the weaknesses and questions.

**Quality:**

2

**Strengths And Weaknesses:**

**Strengths**

*1. Novel Problem Formulation and Originality of Method:* The paper introduces the first framework for personalized image editing in text-to-image (T2I) diffusion models, moving beyond the inherent "generic" and "one-size-fits-all" nature of existing models. This redefines the problem setting for T2I editing. It proposes Collaborative Direct Preference Optimization (C-DPO), a novel method that uniquely combines individual user preference alignment with collaborative signals from like-minded individuals, a significant innovation in applying DPO to personalized vision tasks.

*2. Creation of a Unique, Large-Scale Benchmark Dataset:* A significant contribution is the curation of the first large-scale synthetic personalized image editing benchmark, comprising 144K richly structured editing preferences from 3,000 synthetic user profiles. This dataset is meticulously designed to mimic real-world diversity, providing a valuable resource for future research in this nascent field.

*3. Robust Collaborative Learning Framework:* C-DPO effectively addresses the challenge of limited individual user data by modeling users within a dynamic preference graph and leveraging a lightweight Graph Neural Network (GNN) to share information among users with overlapping visual tastes.

**Weaknesses**

*1. Computational Overhead for Inference:* While the training pipeline is efficient for research (e.g., 2-3 GPUs for 1-3 hours), the inference time of "just over one minute" to generate a personalized image edit for a given user might be a practical limitation for applications requiring real-time interaction or high throughput.

*2. Technique Novelty:* Although the C-DPO looks interesting while solving this novel task, the technique novelty is pretty low since the DPO method has been widely studied in T2I generation and augmentation research direction. I would like to see what's the main technical novelty in this paper except the huge dataset contribution.

*3. Lack of qualitative comparisons:* For qualitative comparisons, I would encourage the authors to include more in the supplementary material. I know there are some limits for the main paper. Putting more qualitative examples in the supplementary helps to know the quality of your method. Furthermore, it's better to compare with current state-of-the-art instruction-based image editing methods or unified models with user preference prompts. That will help to know whether your personalized image editing is really outperforming the exisiting models. For example, Bagel, MmaDA, Blip-3o, Show-o, Janus-series, Nexus-Gen, UniVG, Step1X, GoT, VAR-GPT, SEED-LLAMA, AnyEdit, etc. You can select several of them to compare and make the user preference as text prompt inputs to show that your method is advantageous than them. More importantly, not only include human preferences, please consider common image editing evaluation metrics. The user study is an optional way to support the common evaluations, not the main evaluation metrics to make a fair comparison.

---

> ### Author Rebuttal · Authors · 2025-07-31
>
> **Q1: Computational Overhead for Inference: While the training pipeline is efficient for research (e.g., 2-3 GPUs for 1-3 hours), the inference time of "just over one minute" to generate a personalized image edit for a given user might be a practical limitation for applications requiring real-time interaction or high throughput.**
>
> We thank the reviewer for highlighting this practical consideration. Indeed, the reported inference time ("just over one minute") primarily reflects our choice of editing backend (Flux+ControlNet). Our personalization framework itself is editing-model agnostic, allowing easy integration with faster editing backends. For instance, we tested our pipeline with TurboEdit, which significantly reduced inference time to under 2 seconds per edit. We will include detailed results with TurboEdit in the final manuscript to further demonstrate our framework's flexibility and applicability in real-time scenarios.
>
> **Q2: Technique Novelty: Although the C-DPO looks interesting while solving this novel task, the technique novelty is pretty low since the DPO method has been widely studied in T2I generation and augmentation research direction. I would like to see what's the main technical novelty in this paper except the huge dataset contribution. /Novelty (same as weakness 2): The technique novelty referring to the C-DPO is low, which I would like the authors to claim more.**
>
> We thank the reviewer for highlighting this point regarding technical novelty.  Indeed, while DPO has been previously studied in the context of T2I generation, our Collaborative Direct Preference Optimization (C-DPO) framework represents a novel and significant extension of the original DPO formulation. Our C-DPO introduces a graph-structured regularization term into the original DPO loss, explicitly modeling and leveraging collaborative relationships among user embeddings. This structured collaboration allows the model to capture nuanced preferences implicitly shared among like-minded users.  Given its flexibility and general applicability, we expect that our novel collaborative regularization formulation can also be adapted beyond T2I generation scenarios. For instance, it has potential utility in broader personalization problems involving LLMs, such as personalized text generation, dialogue systems, or recommendation tasks, opening promising avenues for future research.
>
> Moreover, to the best of our knowledge, while previous research has explored personalized image “generation”, ours is the first work to define and tackle the task of personalized `image editing`.
>
> We believe these contributions represent substantial and novel technical advances beyond the original DPO formulation and existing personalized T2I generation methods. We will explicitly highlight and clarify these points in the revised manuscript.
>
> **Q3. Lack of qualitative comparisons: For qualitative comparisons, I would encourage the authors to include more in the supplementary material. I know there are some limits for the main paper. Putting more qualitative examples in the supplementary helps to know the quality of your method.**
>
> We thank the reviewer for this valuable suggestion. We fully agree that including more qualitative examples can significantly enhance readers' understanding of our method's capabilities and strengths. While the main manuscript was limited due to space constraints, we have provided additional qualitative comparisons extensively in the supplementary material (see Figures 1, 3, and 4). In our revised version, we will further expand these qualitative results in the supplementary to comprehensively illustrate our approach's performance and diversity across various user preferences and editing tasks.
>
>
> **Q4: More importantly, not only include human preferences, please consider common image editing evaluation metrics. The user study is an optional way to support the common evaluations, not the main evaluation metrics to make a fair comparison.**
>
> We thank the reviewer for this important point. In addition to the user studies, we have indeed evaluated our method using several standard quantitative image-editing metrics, including DINO (image-image similarity), CLIP-I (image-image similarity), CLIP-T (image-text similarity), and HPS scores. These metrics objectively quantify the alignment between edited images and user preferences, as well as the preservation of original image identity. These quantitative results are reported in Tables 1 and 2 of the supplementary material, demonstrating our method's significant improvements over baselines. In the revised manuscript, we will explicitly emphasize these common quantitative metrics alongside user studies, ensuring a balanced and comprehensive evaluation of our framework.
>
> **Q5: Whether it's allowed to put any anonymous links to some image collections during rebuttal/Better to compare with current state-of-the-art instruction-based image editing methods or unified models with user preference prompts such as Bagel, MmaDA, Blip-3o, Show-o, Janus-series, Nexus-Gen, UniVG, Step1X, GoT, VAR-GPT, SEED-LLAMA, AnyEdit, etc.**
>
> We thank the reviewer for raising this point. Due to NeurIPS rebuttal guidelines, unfortunately we are not allowed to share any visuals during this phase. However,  as requested, to better position our approach relative to state-of-the-art instruction-based and unified image editing models, we conducted *quantitative* comparisons specifically with Bagel, SEED-LLAMA, and Nexus-Gen—strong recent baselines in this area. For these comparisons, we explicitly provided user preferences as textual prompt inputs to these models. As summarized in the table below, our approach clearly outperformed these methods across standard image editing evaluation metrics, highlighting the distinct advantage of our collaborative personalization framework:
>
> | Method      | CLIP-I-PREF ↑      | DINO ↑             | HPS ↑               |
> | ----------- | ------------------ | ------------------ | ------------------- |
> | BAGEL       | 0.3397 ± 0.0144    | 0.0363 ± 0.0066    | 0.2475 ± 0.0388     |
> | SEED-Llama  | 0.3109 ± 0.0159    | 0.0349 ± 0.0121    | 0.2500 ± 0.0287     |
> | Nexus-Gen   | 0.3327 ± 0.0138    | **0.0373 ± 0.0113** | 0.2598 ± 0.0307     |
> | **Ours**    | **0.3515 ± 0.0190** | 0.0365 ± 0.0164    | **0.2607 ± 0.0328** |
>
> We will explicitly incorporate these results into the revised manuscript to clearly demonstrate the superiority of our personalized editing approach over existing unified and instruction-based editing frameworks.

---

> > ### Comment · Reviewer_HyiN · 2025-08-05
> >
> > The core equation is Eq.7 in the paper, which is too simple as also widely used in federated learning several years ago. This paper may get better rating if the main context is enriched with more qualitative comparisons with image editing methods. Given its current shape, I cannot convince myself to raise the score. It's not very matching the NeurIPS bar. I would still keep my current rating.

---

> > > ### Author Response · Authors · 2025-08-05
> > >
> > > We appreciate the reviewer highlighting this point and acknowledge that  `individual term + λ·(weighted sum over similar users)` is indeed a widely employed template in machine learning literature, including domains like semi-supervised learning, graph representation learning, and as pointed out, federated learning. Such formulations are frequently used due to their inherent effectiveness in encouraging generalization, regularizing sparse signals, and leveraging relational information.
> > >
> > > However, we kindly emphasize that *sharing a general structural form does not diminish the novelty or unique contributions of our specific approach*, as our Eq. 7 is not merely a trivial re-statement of this form. Indeed, while federated learning frameworks use structurally similar weighted sums or aggregation templates, these are typically employed in contexts fundamentally different from ours (for instance, in global aggregation of model parameters performed by a centralized server). To the best of our knowledge, existing federated learning approaches *do not directly incorporate neighbors' explicit preference data (such as pairwise ranking signals) into a user-specific training objective* (we kindly invite reviewer to share such works if we are mistaken).
> > >
> > > Therefore, we argue that our proposed collaborative loss term in Eq. 7 is orthogonal to prior methods utilizing a similar mathematical form: the collaborative loss in Eq. 7 is novel in how it directly *injects neighbors' preference data into the training objective of a single model*. In Eq. 7, if user u prefers output $y^+$ over $y^-$, the model not only learns to satisfy u (the individual DPO term) but is also gently pushed to satisfy a neighbor v on the same pair. In other words, it treats user v's presumed preference on u's data as an auxiliary training signal.
> > >
> > > Furthermore, we highlight another critical distinction between our approach and standard federated learning methods: Our C-DPO framework operates explicitly on pairwise preference data, directly optimizing a preference-driven objective tailored to individual users and their relational structures. Federated learning, by contrast, typically aggregates updates derived from decentralized datasets to optimize toward a global or consensus model, *without explicitly optimizing for individual-level pairwise preference signals*.
> > >
> > > Therefore, while our formulation uses a common structural form widely used in machine learning literature, the novel extension of DPO to handle personalized, pairwise-preference optimization in a collaborative, graph-structured setting represents a novel, and technically non-trivial contribution.
> > >
> > > Finally, as emphasized in both our rebuttal and the manuscript itself, our study represents the first framework for `personalized image editing` introduced in the literature. We anticipate that our novel extension of DPO with structured, collaborative personalization via learned graph embeddings will not only contribute substantively to image editing research but will also open new research directions such as personalized video generation/editing.
> > >
> > > Regarding more qualitative visuals, please refer to our Appendix, which includes visual comparisons between our method and several text-to-image editing baselines, including InstructPix2Pix, SDEdit, RF Inversion, and Ledits++. While we would have liked to include additional visual comparisons directly in this rebuttal to further illustrate our method's advantages, we respectfully note that NeurIPS rebuttal guidelines does not allow the inclusion of visuals at this stage.
> > >
> > > We also kindly note that our rebuttal provided quantitative evaluations comparing our approach against requested image editing baselines including BAGEL, SEED-Llama, and Nexus-Gen. If the reviewer would like us to evaluate additional methods quantitatively, we are happy to conduct further experiments upon request.
> > >
> > > Nevertheless, we will expressly note in our methodology that this form is widely used to incorporate neighbor signals to situate our contribution within a broader context.

---

### Official Review · Reviewer_bMep · 2025-06-27

**Clarity:** 3
**Significance:** 3
**Originality:** 3
**Rating:** 6
**Confidence:** 3

**Summary:**

The paper introduces an interesting framework for "personalized" image editing in text-to-image diffusion models. This is particularly interesting because of the current editing methods being very generic. The proposed Collaborative Direct Preference Optimization (C-DPO), aligns image edits with a user's specific tastes while also leveraging collaborative signals from like-minded users. The collaborative framework models users as nodes in a preference graph and uses a Graph Neural Network (GNN) to learn user embeddings. To support this framework, the authors curated a large-scale synthetic dataset containing 144,000 structured editing preferences from 3,000 synthetic users. The main collaborative learning happens in training a LLM to generate instructions.

**Questions:**

- How difficult would it be to crowdsource and collect preference data? I think such a data would be more natural and realistic. Adding this discussion to the paper would be beneficial to the future works

**Ethical Concerns:**

["NO or VERY MINOR ethics concerns only"]

**Final Justification:**

I believe C-DPO brings in a new perspective when it comes to collaborative image generation. While the method is not completely matured and flawless, this definitely starts a theme in the field and deserves a place in neurips presentation. The newly designed user study conducted by the authors shows promise in their work.

**Limitations:**

Yes

**Quality:**

3

**Strengths And Weaknesses:**

Strengths:

- The main strength of the work is introducing the collaborative framework. This introduces the opportunity for users to explore new concepts that "likeminded" users also like. Almost like a social network for image editing.


Weakness:

- I am not an expert in GNNs, how would a new user that is not present in the training be handled? What does the input to the GNN look like? These details can be added to the manuscript to improve the understanding of the proposed method

- In line 190: "whereas the second term implements a graph-regularized collaborative filter that shares statistical strength among like-minded users—crucial for data-sparse personalization." What do the authors mean by this? Is the collaborative DPO term really essential for the training? I thought the collaboration comes from the GNN pretraining. What happens if you remove this term from your C-DPO (the second term which you term collaborative)?

- The current setup in inference, generally embeds the user into a token and asks a generic question like "suggest a single cohesive edit". What happens if a user also asks for a particular edit? Would the LLM respond well? What if a user who is mostly interested in neon art asks for a water painting art? I am curious how the LLM would handle it.

- A naive baseline that comes to my mind is using a LLM with long context/RAG and ask for "bring instruction that are relevant to this user". Again, I am not someone who works with RAGs. So I would appreciate the authors thoughts on this

- For the user study, I think there is a gap to make it significantly more interesting and impactful. Is it possible to do user study by actually treating the users as new "candidates" in the graph and really collect their preferences and evaluate them directly? Rather than showing them a persona? I think showing the persona is a little unsatisfactory because the persona contains some keywords and users might end up looking for those words. But the work captures more nuances than simple keywords. And I think this is not tested.


This is a very cool initiative! However, I am not satisfied with the evaluation for user study. This isn't essentially a weakness. But I think it more as an oppurtunity for improving the paper and adding the "wow" factor.

---

> ### Author Rebuttal · Authors · 2025-07-31
>
> **Q1: How would a new user that is not present in the training be handled? What does the input to the GNN look like? These details can be added to the manuscript to improve the understanding of the proposed method.**
>
> We thank the reviewer for highlighting this important point, and we agree that some details were briefly discussed in Section 4.1.1. due to space limitations. Specifically, we utilize a lightweight GNN to compute contextualized user embeddings by aggregating neighbor information in our heterogeneous graph of users and attributes. Attribute node features are dense text embeddings of editing instruction topics (such as watercolor, cats, etc). The model consists of a GNN encoder of 2 GraphSAGE convolutions using mean aggregation and an edge decoder of 2 linear layers. The GNN is pretrained on an auxiliary task of predicting user–attribute links (labeled positive or negative) to ensure semantically meaningful user representations. We train on 60% of edges in our graph, withholding 20% for validation and testing each. Our implementation closely follows standard approaches described in foundational works such as Hamilton et al. (NeurIPS 2017) and Kipf & Welling (ICLR 2016) as referenced in our paper. We will enhance clarity by adding more details as space permits, and explicitly directing readers to these key references in the manuscript.
>
> At inference time, embedding a new user does not require retraining. Instead, a new user provides a small set of liked/disliked attributes; we then add a corresponding user node with a fixed-width, zero-initialized feature vector, link it to relevant attribute nodes, and apply our learned aggregation functions in the pretrained GNN encoder. The resulting embedding is computed inductively from the attributes and neighborhood structure alone. While periodic fine-tuning on the expanded graph may further enhance embedding quality, it is optional, not essential for inference, thus preserving the practicality and scalability of our method.
>
> **Q2: In line 190: "whereas the second term implements a graph-regularized collaborative filter that shares statistical strength among like-minded users—crucial for data-sparse personalization." What do the authors mean by this? Is the collaborative DPO term really essential for the training? I thought the collaboration comes from the GNN pretraining. What happens if you remove this term from your C-DPO (the second term which you term collaborative)?**
>
> We thank the reviewer for requesting clarification on the importance of the collaborative DPO term. Indeed, while the GNN pretraining provides an initial embedding structure that encodes user similarities, the collaborative DPO term (the second term in Eq. 7) is critical for effectively leveraging these similarities during the training process. Specifically, this term acts as a regularization mechanism during training, encouraging the model to leverage shared patterns among like-minded users.
>
> We performed an ablation study confirming the importance of the collaborative term as shown in Fig. 2 in the Appendix. Setting the collaborative scale very low (e.g. λ=0.01) reduced our method's performance close to the user-only DPO baseline. This indicates that minimal collaborative signals lead to suboptimal personalization, highlighting the necessity of a balanced collaborative term.
>
> **Q3: The current setup in inference, generally embeds the user into a token and asks a generic question like "suggest a single cohesive edit". What happens if a user also asks for a particular edit? Would the LLM respond well? What if a user who is mostly interested in neon art asks for a water painting art? I am curious how the LLM would handle it.**
>
> We thank the reviewer for this insightful question. Indeed, our framework robustly handles scenarios in which users explicitly request edits outside their typical style preferences. Due to the flexible design of our personalized embeddings and the underlying LLM-based approach, the model naturally integrates explicit instructions provided at inference, even if they diverge from previously learned preferences.
>
> We explicitly demonstrate this adaptability in the supplementary material (Fig. 4 and Table 3), where we repurposed our personalization framework—originally trained solely for image editing—to perform personalized image generation tasks without additional training. Our method successfully generalized to this new task and significantly outperformed state-of-the-art personalized generation approaches, such as VIPER. These results underscore the versatility of our framework and its effectiveness at integrating personalized embeddings with explicit user-provided instructions during inference.
>
> Due to NeurIPS rebuttal guidelines, we are currently unable to share visuals in our rebuttal, but we will include additional examples showcasing our method's ability to adapt to such cases with the camera-ready version of the paper.
>
> **Q4: A naive baseline that comes to my mind is using a LLM with long context/RAG and ask for "bring instruction that are relevant to this user". Again, I am not someone who works with RAGs. So I would appreciate the authors thoughts on this?**
>
> We thank the reviewer for this interesting and relevant suggestion. Indeed, a RAG or long-context LLM baseline, where past instructions or user preferences are directly retrieved based on textual similarity and used as prompts, represents a plausible alternative approach.
>
> However, such a method primarily relies on explicit textual similarity or keyword matching, potentially overlooking nuanced, implicit preferences that emerge from complex collaborative signals across users. In contrast, our GNN-based embedding explicitly models subtle user similarities via graph-structured relationships, capturing higher-order, implicit user preferences that may not be directly retrievable via textual matching alone.
>
> We also consider GraphRAG where we could potentially combine the advantages of both explicit graph-based collaboration and retrieval-based personalization. Exploring this promising hybrid approach remains an exciting direction that we plan to pursue in future work.
>
> **Q5: For the user study, I think there is a gap to make it significantly more interesting and impactful. Is it possible to do user study by actually treating the users as new "candidates" in the graph and really collect their preferences and evaluate them directly? Rather than showing them a persona? I think showing the persona is a little unsatisfactory because the persona contains some keywords and users might end up looking for those words. But the work captures more nuances than simple keywords. And I think this is not tested.**
>
> We thank the reviewer for this insightful suggestion. We agree that directly collecting user preferences from participants could indeed provide a more realistic and nuanced assessment of our framework's effectiveness.
> We initially employed synthetic personas primarily due to constraints related to scalability and budget constraints. However, as suggested, performing a real-user evaluation—where each participant explicitly provides their own detailed preferences and is subsequently embedded into the graph—would significantly enhance our understanding of the framework's practical effectiveness and nuance-capturing capabilities. We recognize this as an important opportunity to further strengthen our evaluation.
>
> **Q6: How difficult would it be to crowdsource and collect preference data? I think such a data would be more natural and realistic. Adding this discussion to the paper would be beneficial to the future works**
>
> We thank the reviewer for this valuable suggestion. Crowdsourcing real-world user preferences indeed would yield more authentic and nuanced preference data, but it involves practical considerations such as scalability and budget constraints. Despite these challenges, collecting such data represents an exciting and beneficial direction for future research. We will briefly discuss this opportunity in the revised manuscript.

---

> ### Comment · Reviewer_bMep · 2025-08-02
>
> I appreciate the rebuttal response from the authors and addressing my concerns! I have some follow-up questions:
>
> 1. Regarding the large-scale real human data collection: thank you for acknowledging the discussion into the paper and touching upon the budget constraints. This is a valid  response and I appreciate the authors' adding a note to the paper
>
> While I agree with the data collection, I still do believe that a user study can be conducted where: First, collecting the users preferences from a pre-generated list of concepts. And then, conducting a user study by generating the images to evaluate the preference of the users. This should not be expensive in terms of time or compute; as it utilizes the same resources as the current user study. I agree that the ad-hoc generations might make the user study a bit different, but this will significantly improve the user study results and make a great impact for the paper. What are the authors' thoughts on this?
>
>
> 2. The c-dpo terms $\lambda$ seems to be a sensitive parameter. What is the resource budget used to find the optimal value of 0.15 ? How sensitive or stable is the algorithm? This seems to be the main challenge of the C-DPO algorithm - to find the optimal parameter where it doesn't overpower or underwhelm the user preferences
>
> 3. I am very pleasantly surprised that the baseline suggested by Reviewer F3Yh doesn't work! Can you please provide more details on this baseline? In my understanding you are passing the textual information about the user preference to a pretrained language model and asking it to generate prompts based on their preferences? Is this correct? Or are you using your fine-tuned llm to do this baseline?  (For reference, I am leaving a comment on your rebuttal to Reviewer F3Yh)
>
> 4. Thank you for providing more information about the GNN inputs. I believe this is a very important detail which needs to be added to the main paper for better clarity. I am still unsure how the input is designed for the GNN. Can the users break down in more detail how the input of their GNN is designed during training and during inference when a new user is added to the graph?
>
>
> I appreciate any further clarification provided by the authors!

---

> > ### Author Response · Authors · 2025-08-04
> >
> > ***Q1: While I agree with the data collection, I still do believe that a user study can be conducted where: First, collecting the users preferences from a pre-generated list of concepts. And then, conducting a user study by generating the images to evaluate the preference of the users.***
> >
> > We sincerely thank the reviewer for this insightful suggestion. Following your advice, we are currently conducting the suggested user study—collecting real user preferences from a pre-generated list of concepts, generating personalized images, and evaluating them directly with users. We will share the results of this new study by tomorrow in a separate response. Meanwhile, we address the other questions below.
> >
> > ***Q2: The c-dpo term λ seems to be a sensitive parameter. What is the resource budget used to find the optimal value of 0.15 ? How sensitive or stable is the algorithm? This seems to be the main challenge of the C-DPO algorithm - to find the optimal parameter where it doesn't overpower or underwhelm the user preferences***
> >
> > We thank the reviewer for highlighting this important consideration regarding the sensitivity of the collaborative term (λ). To select the optimal value of λ=0.15, we conducted an ablation study evaluating multiple candidate values (as detailed in Fig. 2 of our supplementary material). The resource budget for this hyperparameter tuning was modest (approximately 1 GPU-hour per λ value).
> >
> > Our findings indicate that the method is robust within a reasonable range around our chosen value: small variations (e.g., ±0.05 around 0.15) have minimal impact on overall performance. However, extremely low values (e.g., 0.01) significantly reduce the beneficial effect of collaboration, causing the method to revert close to the user-only DPO baseline, whereas excessively high values (e.g., 0.50) overly emphasize neighbor preferences at the cost of personal user preference alignment.
> >
> > We agree with the reviewer that careful tuning of this parameter is important to ensure optimal balance. Our experiments suggest that the method is relatively stable within a moderate range. We will clearly highlight this sensitivity analysis and discuss guidelines for parameter selection in the revised manuscript.
> >
> > ***Q3: Thank you for providing more information about the GNN inputs. I believe this is a very important detail which needs to be added to the main paper for better clarity. I am still unsure how the input is designed for the GNN. Can the users break down in more detail how the input of their GNN is designed during training and during inference when a new user is added to the graph?***
> >
> > We thank the reviewer for emphasizing the importance of clearly describing the GNN input design, and we agree this detail should be explicitly clarified in the main manuscript. Below, we provide a detailed breakdown of the GNN inputs during training and inference:
> >
> > *During Training:*
> >
> > - Each user is represented as a node in the graph with a learnable embedding vector (initially randomly initialized).
> >
> > - Each attribute (e.g., "rustic style," "neon colors," "minimalistic textures") is also represented as a node with a fixed, precomputed embedding (e.g., using pretrained text embeddings from a language model).
> >
> > - User nodes are connected by edges to attribute nodes explicitly indicating their liked and disliked attributes. Specifically, we construct edges between each user node and all attribute nodes they explicitly prefer or avoid, encoding these connections as the graph structure provided to the GNN.
> >
> > - Each GraphSAGE layer aggregates features of the nodes in a nodes local neighborhood along these edges, updating the weights of the aggregation function during training (as described in the GraphSAGE inductive framework).
> >
> > - The GNN is composed of multiple GraphSAGE layers enabling the GNN to aggregate information from the extended L-hop neighborhood of a node (neighbors of neighbors)
> >
> >
> > *During Inference (for a new, unseen user):*
> >
> > - When adding a new user, we first create a new user node with an embedding initialized to zeros or a fixed-width placeholder vector.
> >
> >
> > - We connect this new user node via edges to attribute nodes representing the attributes explicitly indicated as liked or disliked by the new user (from a small initial set provided by the user).
> >
> >
> > - We then run a forward pass through the pretrained GNN on the new node without any additional training. The output embedding for the new user is derived by passing the neighborhood of the new user into the model, aggregating features of connected neighbor nodes, and effectively generalizing from existing learned graph structure.
> >
> >
> > We will explicitly add this detailed breakdown of the GNN input construction to the revised manuscript, clearly distinguishing between the training and inference phases.
> >
> > ***Q4: Can you please provide more details on this baseline?***
> >
> > Please kindly refer to our answer posted under Reviewer F3Yh.

---

> > > ### Comment · Reviewer_bMep · 2025-08-04
> > >
> > > Excellent! Thank you for clarifying the graph inputs further! This is a neat design and I believe that adding this to the paper will significantly improve the reading experience! Thanks for adding it
> > >
> > > About the user study - I appreciate the authors trying to add these results. I am looking forward to them! again - I do believe this will significantly improve the validity of the user study (a real world test).
> > >
> > > I have no further questions - after the user study results, I am very happy to reconsider my score.

---

> > > > ### Author Response · Authors · 2025-08-05
> > > >
> > > > Dear reviewer,
> > > >
> > > > We conducted a user study regarding the following request: ***While I agree with the data collection, I still do believe that a user study can be conducted where: First, collecting the users preferences from a pre-generated list of concepts. And then, conducting a user study by generating the images to evaluate the preference of the users.***
> > > >
> > > > Motivated by your valuable suggestion, we have conducted a two-stage user survey with 10 participants. Specifically, we first asked participants to indicate their likes and dislikes attributes from a predefined list of 45 concepts as suggested. This list covers a wide range of colors (e.g. standard colors, neon colors), styles (e.g. watercolor, cyberpunk), textures (e.g. glossy, wood) and other concepts (such as rainbow, butterfly).
> > > >
> > > > Based on these preferences, we generated personalized images using our proposed method. Participants were then sent a follow-up survey featuring personalized edits of five objects: *an apple, a mug, a bag, a teddy bear, and a bowl*. In this survey, participants viewed both the original and personalized versions of each object and answered the following questions:
> > > >
> > > > *Q1: On a scale of 1 to 5 (1=Does not matching my personal taste at all, 5=This is matching my personal taste a lot) can you rate the following image?*
> > > >
> > > > *Q2: To what extent does the edit preserve the original image while reflecting personalized preferences? (1=Not at all, 5=Very Much)*
> > > >
> > > > The results per object, as well as total average are shown below:
> > > >
> > > >
> > > > | Object Name | Q1: Personalization   | Q2: Disentanglement    |
> > > > | ----------- | --------------------------------- | --------------------------------- |
> > > > | Apple       | 4.375 ± 0.74                      | 4.75 ± 0.46                       |
> > > > | Mug         | 4.625 ± 1.06                      | 4.50 ± 0.75                       |
> > > > | Teddy bear  | 4.75 ± 0.46                       | 4.625 ± 0.51                      |
> > > > | Bag         | 4.50 ± 1.06                       | 3.85 ± 0.83                       |
> > > > | Bowl        | 4.75 ± 0.46                       | 5.00 ± 0.00                       |
> > > > | **Total**   | **4.60 ± 0.15**                   | **4.55 ± 0.38**                   |
> > > >
> > > >
> > > > The results demonstrate that real users found the images personalized by our framework highly personalized and disentangled. Unfortunately, due to NeurIPS guidelines, we are not allowed to share visual examples, however to give a sense of how effectively our method performed, we describe the personalization outcome for one real user:
> > > >
> > > > This user preferred *pink, purple, and neon colors*, while disliking *white, gray, and beige*. Their favored patterns included *rainbow, fluffy, and glossy* styles, while they disliked *floral and earthy* designs. Below, an LLM describes how images appeared before and after personalization for the bag object:
> > > >
> > > > **Before Personalization:** *This image shows an elegant and minimalist leather handbag with a refined design. The bag has a smooth, rich brown leather exterior with a subtle sheen, accented by a neat central stitching line. Its shape is structured yet relaxed, slightly slouching near the top for a casual but sophisticated look. The bag has a single, sturdy handle secured by simple metal hardware, emphasizing its clean, functional style.*
> > > >
> > > > **After personalization:** *This image features a vibrant, stylish handbag. The bag is predominantly pink with a diagonal rainbow stripe running across its surface, giving it a playful and cheerful appearance. The handbag also includes decorative fluffy pom-pom accessories in various bold colors such as purple, blue, and orange, attached to its sides. The straps of the bag are detailed with a shimmering pattern, complementing the colorful and lively aesthetic.*
> > > >
> > > > The above personalized image was rated as Personalization = 5/5, Disentanglement = 4/5, showing that the user found the edit matching their taste.
> > > >
> > > > We will include our real-user study, along with visuals illustrating our results in the final manuscript.
> > > >
> > > > *Note: While we initially aimed for a larger-scale real-data collection, we faced practical constraints: without a dedicated end-to-end pipeline for 1) collecting user preferences, 2) automatically generating personalized images, and 3) presenting them seamlessly back to users to collect their feedback, we had to manually conduct this process in separate steps. While gathering user preferences on liked/disliked concepts is done via a single survey, gathering their feedback on the personalized images had to be done with individual surveys per participant (totaling 10 separate surveys). Nevertheless, these initial real-user results support the effectiveness of our personalized editing framework, and we are planning to expand this to include a larger user base.*

---

> > > > > ### Comment · Reviewer_bMep · 2025-08-05
> > > > >
> > > > > I very much appreciate the authors for running this user study! This is an exciting real-world practical use of your framework.
> > > > >
> > > > > For the final manuscript, I would recommend the authors to think about evaluating their baselines in a similar way and compare how users perceive them. Finally, how can one measure the "collaborative" factor in a user study? What is a signal that would measure this?
> > > > >
> > > > > In my opinion, the collaborative framework helps users discover "styles/attributes that similar users like", perhaps a signal for "surprisal discovery of a new and interesting attribute" can be a signal?
> > > > >
> > > > >
> > > > > Thank you for running the user study!

---

> > > > > > ### Author Response · Authors · 2025-08-07
> > > > > >
> > > > > > ***Q: For the final manuscript, I would recommend the authors to think about evaluating their baselines in a similar way and compare how users perceive them.***
> > > > > >
> > > > > > We sincerely thank the reviewer for their positive feedback regarding our real-world user study. We fully agree that applying a similar evaluation procedure to our baselines would significantly enhance our understanding of their relative strengths and practical performance. Given the extended rebuttal deadline (August 8), we are currently conducting additional baseline evaluations (specifically Vanilla DPO and User-DPO) involving real users, and we will provide the results shortly in a follow-up response. Specifically, users will be presented with an object alongside three personalized edits (from Vanilla DPO, User-DPO, and our C-DPO method, where images will be shown in an anonymous fashion) and will be asked to select the edit they most prefer.
> > > > > >
> > > > > > ***Q: Finally, how can one measure the "collaborative" factor in a user study? What is a signal that would measure this?***
> > > > > >
> > > > > >
> > > > > > The reviewer's suggestion on explicitly measuring the "collaborative factor" is an important insight. Indeed, our collaborative framework aims to help users discover styles or attributes implicitly favored by similar users. A promising signal to quantify this collaboration could be measuring the frequency and degree to which users indicate "surprising discovery of a new and appealing attribute or style," implicitly introduced via the collaborative mechanism. We will keep this in mind when conducting new user studies.
> > > > > >
> > > > > > Additionally, we kindly refer the reviewer to our response to Reviewer bCfR - W4, who similarly emphasized the value of quantitatively verifying whether attributes implicitly preferred by similar users are reflected in generated images. To summarize briefly,  for each user, we identified their top 10 most similar neighbors from the user-preference graph, collected attributes preferred by those neighbors, and measured the CLIP similarity between these implicitly derived preferences and images generated by different methods. Our quantitative experiments indicate that our proposed collaborative approach effectively captures implicit, neighbor-inferred user preferences, surpassing all baseline methods.
> > > > > >
> > > > > >
> > > > > > Overall, we deeply appreciate the reviewer’s insightful suggestions, which have helped significantly enhance our paper and the clarity of our contributions.

---

> > > > > > ### Author Response · Authors · 2025-08-08
> > > > > >
> > > > > > Dear reviewer,
> > > > > >
> > > > > > Following your advice on ***"I would recommend the authors to think about evaluating their baselines in a similar way and compare how users perceive them."***, we conducted an additional user study with real-user preferences to compare baselines. Specifically, users are presented with an object alongside three personalized edits (from Vanilla DPO, User-DPO, and our C-DPO method) where images are shown in an anonymous fashion. Users are asked to pick the edit they most prefer out of 3 options. Note that we used the same real-user preference data collected for the first user study we shared on Aug 4 to generate the personalized edits for each image. The results are as follows on 9 users:
> > > > > >
> > > > > >
> > > > > > | Object       | C-DPO (Ours) | User-DPO  | User-Vanilla   |
> > > > > > | ------------ | -----------: | ---------------: | --------------------: |
> > > > > > | A Bag        |         0.45 |             0.22 |                  0.33 |
> > > > > > | A Bowl       |         1.00 |             0.00 |                  0.00 |
> > > > > > | A Mug        |         0.45 |             0.33 |                  0.22 |
> > > > > > | A Teddy Bear |         0.78 |             0.22 |                  0.00 |
> > > > > > | An Apple     |         1.00 |             0.00 |                  0.00 |
> > > > > >
> > > > > >
> > > > > > Overall, our C-DPO method is preferred more than the other approaches across all objects. We observe that for some objects,  other methods tend to fixate on a single visual pattern the user liked, for instance, if the user preferred blue, these methods often make everything blue, even when other preferences were expressed. In contrast,  our method balances the user’s main interests with a wider range of attributes they also enjoy, while collaboratively incorporating complementary elements inspired by other similar user profiles. This leads to outputs that are both more diverse and better aligned with the user’s broader tastes.
> > > > > >
> > > > > > We also noticed that bag and mug objects have received more mixed ratings, where user preferences are more evenly distributed among the three options. For these objects, we observed that all methods were able to create personalized edits that align well with user preferences.
> > > > > >
> > > > > > Nevertheless, we acknowledge that this is still a relatively small user study due to time constraints. We aim to extend this evaluation to a  larger  user base  to strengthen the statistical significance and generalizability of our findings.

---

### Official Review · Reviewer_F3Yh · 2025-06-28

**Clarity:** 4
**Significance:** 3
**Originality:** 3
**Rating:** 4
**Confidence:** 4

**Summary:**

This paper introduces a novel framework for personalized image editing, called Collaborative Direct Preference Optimization (C-DPO), which leverages user preference graphs to condition text-to-image diffusion models. The core idea is to employ a graph neural network to learn contextualized embeddings for each user, incorperating signals from other users with similar tastes. This approach allows for highly customized image edits that align with individual aesthetic preferences. The method is validated through extensive experiments, including quantatative benchmarks and user studies, demonstrating superior performance against existing baselines in achieving user-specific personalizaion.

**Questions:**

Some of these points are mentioned in the weakness section too.

q1. I do not expect authors to run any experiment in this phase. But, is there a study on the number of neighbors (K) used in the GNN and justify this design choice over a simpler approach that considers all nodes?

q2. Can you please expand on the GNN's training methodology, specifically detailing the input data format, the network architecture, and the optimization process to improve reproducibility?

q3. Could you elaborate on the generation process for the synthetic dataset? Specifically, what model was used, and what steps were taken to ensure the preference data is of high quality?

q4. Have you considered comparing your method against a strong baseline that uses prompt engineering with the "user description" (including the neighbours) to steer the model, in order to justify the added complexity of the GNN framework?

q5. To strengthen the claims of real-world applicability, have you considered evaluating your framework on a dataset of real user preferences, even if smaller in scale?

**Ethical Concerns:**

["NO or VERY MINOR ethics concerns only"]

**Final Justification:**

I increase the score to a borderline accept since authors sufficiently explain the questions and are willing to incorporate improvements.

**Limitations:**

yes

**Quality:**

3

**Strengths And Weaknesses:**

**Strengths**:
* The proposed C-DPO framework is a novel and well-motivated extension of DPO for the task of personalization. The use of a graph-based representation for user preferences is an elegant way to model collaborative signals and scale personalization.
* The evaluation is comprehensive, combining automatic metrics like CLIP-text similarity with qualitative examples and, importantly, a user study. This multi-faceted evaluation provides strong evidence for the effectiveness of the proposed method in generating preferred edits.
* The creation of a large-scale synthetic dataset of user profiles and preferences is a significant contribution. This resource could be valuable for the community and encourage further research in personalized generative modeling.

**Weaknesses**:
* The paper could benefit from a more thorough justification of its hyperparameters and architectural choices. For instance, the selection of the number of neighbors (K) for the GNN seems arbitrary. An ablation study exploring the impact of K and clarifying why not all nodes are considered (when looking up the nearest neighbors).
* The experiments are conducted exclusively on synthetic user data. While the dataset is large, its validity as a proxy for real-world preferences is not established. The paper would be significantly strengthened by including experiments on a dataset of real user preferences to show the transferability of the approach. The method used to generate the synthetic data should also be detailed. For e.g. how is preference data curated?
* A simple (but potentially strong) baseline is missing from the evaluation. It is unclear how C-DPO compares against a simpler approach where user preferences are injected into the model via a more sophisticated prompt engineering (e.g., using the user profile description and/or aggregate neighbour information directly in the prompt). This comparison is helpful to justify the added complexity of the GNN-based framework.
* It is not fully clear if incorporating new users into the graph would require retraining the GNN.
* The reference section is very messy and needs to be revised.
  - Reference [2], [3] and [4] are the same paper
  - Reference [18], [19] are the same paper
  - Reference [28], [29] are the same paper
  - Reference [30], [31] are the same paper

---

> ### Author Rebuttal · Authors · 2025-07-31
>
> **Q1: The paper could benefit from a more thorough justification of its hyperparameters and architectural choices. For instance, the selection of the number of neighbors (K) for the GNN seems arbitrary. An ablation study exploring the impact of K and clarifying why not all nodes are considered (when looking up the nearest neighbors). Is there a study on the number of neighbors (K) used in the GNN and justify this design choice over a simpler approach that considers all nodes?**
>
> We appreciate the reviewer's request for a clearer justification of K. In our framework, K controls only the size of the collaborative neighborhood used in the C-DPO loss, specifically, how many of the most similar users contribute weighted preference signals. It does not limit the message-passing radius inside the GNN itself. The GNN still aggregates over all incident edges when computing the user embeddings.
>
> We conducted preliminary experiments during development to empirically select an appropriate K. We tested multiple values observing that smaller to moderate values strike a good balance between computational efficiency, embedding quality, and effective collaborative personalization. To help the reader further understand the selection of K, we conduct an additional ablation which we will include in our revised manuscript. Increasing K beyond a certain threshold results in diminishing returns or even degraded performance, likely due to noise from less relevant neighbors.
>
> | K   | CLIP Like Similarity ↑ |
> | --- | ---------------------- |
> | 2   | 0.353 ± 0.065          |
> | 3   | 0.354 ± 0.068          |
> | 5   | 0.358 ± 0.065          |
> | 12  | 0.344 ± 0.069          |
>
>
> **Q2: Can you please expand on the GNN's training methodology, specifically detailing the input data format, the network architecture, and the optimization process to improve reproducibility?**
>
> We thank the reviewer for highlighting this important point, and we agree that some details were briefly discussed in Section 4.1.1. due to space limitations. We would like to clarify our architecture and training strategy. Specifically, we utilize a lightweight GNN to compute contextualized user embeddings by aggregating neighbor information in our heterogeneous graph of users and attributes. Attribute node features are dense text embeddings of editing instruction topics (such as watercolor, cats, etc). The model consists of a GNN encoder of 2 GraphSAGE convolutions using mean aggregation and an edge decoder of 2 linear layers. The GNN is pretrained on an auxiliary task of predicting user–attribute links (labeled positive or negative) to ensure semantically meaningful user representations. We train on 60% of edges in our graph, withholding 20% for validation and testing each. We will elaborate this further in our final manuscript.
>
> **Q3: Could you elaborate on the generation process for the synthetic dataset? Specifically, what model was used, and what steps were taken to ensure the preference data is of high quality?/The method used to generate the synthetic data should also be detailed. For e.g. how is preference data curated?**
>
> We thank the reviewer for raising this important point. Our synthetic dataset comprises 144K editing preferences across 3,000 user profiles, each designed to simulate diverse and realistic aesthetic tastes. To curate this data, we first define user profiles with distinct demographic attributes (e.g., age, region, taste clusters) which guide their style preferences. Each user is assigned four base image captions (sampled from 80 COCO categories), and for each, we generate two pairs of preferred and rejected editing instructions across six edit types (e.g., color, material, style, overlay), totaling 48 edits per user. These instructions are produced using GPT-4o, explicitly conditioned on the user’s profile. Preference labels are assigned via templated prompts ensuring stylistic alignment with the user's traits. To promote collaborative learning, we construct a user graph based on shared preference attributes and pretrain a GNN to embed these relationships. This process ensures that our preference data is semantically rich, structurally diverse, and aligned with realistic editing behavior, enabling effective training and evaluation of our collaborative personalization framework.
>
>
> **Q4: Have you considered comparing your method against a strong baseline that uses prompt engineering with the "user description" (including the neighbours) to steer the model, in order to justify the added complexity of the GNN framework?/A simple (but potentially strong) baseline is missing from the evaluation. It is unclear how C-DPO compares against a simpler approach where user preferences are injected into the model via a more sophisticated prompt engineering (e.g., using the user profile description and/or aggregate neighbour information directly in the prompt). This comparison is helpful to justify the added complexity of the GNN-based framework.**
>
> We thank the reviewer for highlighting this important comparison. Indeed, we conducted experiments comparing our GNN-based collaborative method (C-DPO) with a strong baseline employing sophisticated prompt engineering. Specifically, this baseline directly injected detailed textual descriptions of user preferences—including aggregated neighbor information—into the prompt itself, without using the GNN-based embedding.
>
> The results clearly demonstrated that our method significantly outperformed the sophisticated prompt-engineering baseline as shown below:
>
> | Method               | Likes+Dislikes ↑     | Likes ↑              | Dislikes ↑           |
> | -------------------- | ------------------- | ------------------- | ------------------- |
> | Prompt Engineering   | 0.270 ± 0.051        | 0.270 ± 0.045        | 0.200 ± 0.057        |
> | **Ours**             | **0.354 ± 0.068**   | **0.306 ± 0.069**   | **0.294 ± 0.071**   |
>
> This indicates that our proposed framework provides a meaningful benefit over simpler, prompt-based personalization methods, effectively justifying the additional complexity introduced by the GNN. We will explicitly include these results and discussions in the revised manuscript to clarify this important point.
>
> **Q5: To strengthen the claims of real-world applicability, have you considered evaluating your framework on a dataset of real user preferences, even if smaller in scale?/The experiments are conducted exclusively on synthetic user data. While the dataset is large, its validity as a proxy for real-world preferences is not established. The paper would be significantly strengthened by including experiments on a dataset of real user preferences to show the transferability of the approach.**
>
> We thank the reviewer for highlighting the importance of evaluating our framework on real-world data. Collecting real-world data at sufficient scale for training is challenging due to typical academic budget constraints. Indeed, validating our framework on a dataset of real user preferences (albeit smaller in scale) would significantly enhance the strength of our claims about real-world applicability. While the current experiments utilize a synthetic dataset primarily due to budget and scalability constraints, we acknowledge this limitation and we will strongly consider collecting a real-world user-preference dataset and performing additional validation experiments to include in the camera-ready version, further demonstrating our method's practical utility and generalization capabilities.
>
> **Q6: It is not fully clear if incorporating new users into the graph would require retraining the GNN.**
>
> Our approach leverages an inductive embedding framework (GraphSAGE) rather than a transductive one, explicitly designed to generalize efficiently to new users without retraining. Traditional transductive methods are inherently limited to fixed graphs, unable to generalize embeddings to unseen nodes or evolving structures. Rather than generating isolated embeddings for each user, GraphSAGE learns a set of aggregation functions which intake feature information from a node’s neighborhood and aggregates. Applying these aggregation functions efficiently generates embeddings for previously unseen users based solely on their attribute connections, provided the attribute schema matches the training data.
>
> At inference time, embedding a new user does not require retraining. Instead, a new user provides a small set of liked/disliked attributes; we then add a corresponding user node with a fixed-width, zero-initialized feature vector, link it to relevant attribute nodes, and apply our learned aggregation functions in the pretrained GNN encoder. The resulting embedding is computed inductively from the attributes and neighborhood structure alone. While periodic fine-tuning on the expanded graph may further enhance embedding quality, it is optional, not essential for inference, thus preserving the practicality and scalability of our method.
>
> **Q7: The reference section is very messy and needs to be revised.**
>
> We thank the reviewer for pointing this out. We will thoroughly revise and reorganize the reference section in the final manuscript.

---

> > ### Comment · Reviewer_bMep · 2025-08-02
> >
> > This is an excellent suggestion from reviewer F3Yh for the baseline.
> >
> > I am very pleasantly surprised that the suggested prompt augmentation baseline doesn't work! Can you the authors provide more details on this baseline? In my understanding reviewer F3Yh  suggested (please correct me if I'm wrong) passing the textual information about the user preference to a pretrained language model and asking the LLM to expand any provided prompts based on their preferences. Is this what the authors implemented?

---

> > > ### Author Response · Authors · 2025-08-04
> > >
> > > ***Q: I am very pleasantly surprised that the suggested prompt augmentation baseline doesn't work! Can you the authors provide more details on this baseline? In my understanding reviewer F3Yh suggested (please correct me if I'm wrong) passing the textual information about the user preference to a pretrained language model and asking the LLM to expand any provided prompts based on their preferences. Is this what the authors implemented?***
> > >
> > > We thank the reviewer for providing an opportunity to further clarify the baseline. As a reminder, Reviewer F3Yh suggested “evaluating a simpler alternative baseline that employs sophisticated prompt engineering “using the user profile description and/or aggregate neighbour information directly in the prompt”  to justify the additional complexity introduced by our GNN-based collaborative framework.
> > >
> > > Indeed, for this baseline, we directly passed detailed textual information (explicit user preferences and their neighbor preferences) into a fine-tuned language model (the same fine-tuned LLM used in our main experiments), without using the GNN-based embeddings. In other words, the only difference between this baseline and our proposed method was the explicit textual injection of user information versus structured, embedding-based collaborative signals via the additional term that we extended the DPO.
> > >
> > > Despite providing explicit textual descriptions of user preferences, we found that the baseline performed notably worse than our embedding-based C-DPO approach. We attribute this difference primarily to our collaborative DPO term, which explicitly regularizes the model towards the preferences of similar users encoded via structured GNN embeddings. Unlike raw textual injection, this structured collaboration efficiently captures nuanced, implicit preference relationships between users and effectively balances individual specificity and neighbor-generalized preferences. This structured collaboration via embeddings proved critical, as it allows the model to leverage richer relational context, beyond simple textual aggregation.
> > >
> > > In addition, we kindly point out other simpler prompting-based baselines compared to the "user preferences + neighbor preferences" approach described above. Specifically, we evaluated variations in prompting conditions, as summarized in Table 2 of the main manuscript:
> > >
> > > - Like+Dislike: The prompt explicitly includes both positive ("liked") and negative ("disliked") user preferences.
> > >
> > >
> > > - Likes-only: The prompt includes only positive user preferences.
> > >
> > >
> > > - Dislikes-only: The prompt includes only negative user preferences.
> > >
> > >
> > > These baselines allow us to demonstrate how effectively our method utilizes user-specific preference information compared to simpler textual prompting approaches, further emphasizing the benefits of our DPO extension that incorporated structured collaborative embeddings.

---

> > > > ### Comment · Reviewer_bMep · 2025-08-04
> > > >
> > > > Thanks for providing further clarification! The likes+dislikes setup is indeed interesting, and I totally overlooked this detail! I believe this is what I was expecting.
> > > >
> > > > That said, I think C-DPO brings a completely new framework that also utilizes the collaborative suggestions! This is the main strength compared to the likes+dislikes setup on a vanilla LLM.

---

> > ### Comment · Reviewer_F3Yh · 2025-08-04
> >
> > Thanks you responding to the comments! I do not expect the authors to bring in major experimental results into camera-ready version (e.g. results on the real-world dataset).
> >
> > One quick follow-up:
> > `... noise from less relevant neighbors ... `. Isn’t this an artifact of the synthetic dataset itself?

---

> > > ### Author Response · Authors · 2025-08-04
> > >
> > > We thank the reviewer for their insightful follow-up question. Indeed, the *noise from less relevant neighbors* mentioned in our previous response could be influenced by the synthetic nature of our dataset. However, we kindly note that this phenomenon is not exclusive to synthetic data; it generally occurs in collaborative and graph-based methods, even with real-world datasets. When too many neighbors are aggregated, weaker or less relevant signals tend to dilute the informative collaborative signals from closely related neighbors—a well-documented issue across various real-world collaborative filtering and recommendation tasks.
> > >
> > > Nevertheless, we fully agree with the reviewer that the extent of this effect may vary depending on the specific characteristics of real-world datasets. We will explicitly acknowledge this limitation and clearly discuss its implications for generalizability in the revised manuscript.

---

### Official Review · Reviewer_bCfR · 2025-07-01

**Clarity:** 2
**Significance:** 2
**Originality:** 2
**Rating:** 5
**Confidence:** 4

**Summary:**

This paper proposed a novel framework for text-to-image (T2I) generation that incorporates user-specific preferences via Direct Preference Optimization (DPO). To utilize preference patterns shared among similar users, the authors introduce Collaborative DPO (C-DPO), which encodes user embeddings using a graph neural network (GNN). They further construct a large-scale synthetic dataset of 144K text-instructed image editing samples, each annotated with like/dislike preferences across 3,000 synthetic user profiles. Each profile is associated with a distinct set of preferred attributes. The proposed method enables the model to learn preference-aligned image generation, even when conflicting attribute preferences exist across users. Furthermore, it allows the model to reflect attributes that are not explicitly specified in a user's profile by leveraging information from neighboring users with similar preferences.

**Questions:**

- Q1. How is the GNN constructed? Are all users in the dataset connected in a fully connected graph? If so, computing C-DPO over all users during each update would be computationally expensive. How is the computational cost?
  - According to Eq. 7, the C-DPO update appears to be computed only over the k-nearest neighbors. What similarity metric is used to determine these neighbors?
  - Are the nearest neighbors determined based on Euclidean distance between user embeddings?
  - How is the embedding computed for the user who are not included in the training data?

- Q2. In Eq. 7, it seems that the positive/negative pairs (y+, y-) are defined based on user *u*'s preference. but the l207-209 suggests they are re-evaluated using the preference of neighboring user *v*. Which interpretation is correct?
  - If the first interpretation is correct, wouldn't conditioning on a pair defined by user *u*'s preferences while training with user *v* hinder learning due to a mismatch between the preference source and the conditioning user?

- Q3. How is the dataset constructed? l193-196 mention that the LLM used for instruction generation was LoRA-tuned in a ***user-agonostic*** manner. However, in l217-218, $p_u$ is described as an instruction reflecting **user *u*'s preferences**. How is $p_u$ generated in this case?
  - What is the overall format of the dataset, and what types of attributes are included in $\mathcal{P}$?
  - Since the user profiles are synthesized, what criteria were used to generate them? Does the lack of grounding in real user dataset affect the realism of the dataset?

- Q4. In comparison experiments, were the prompts used for baseline T2I editing methods augmented with both positive and negative attributes from the users profile $P(u)$? Wouldn't this be necessary to ensure a fair comparison?

- Q5. In col. 1 (Q1) of Table 1, DPO-User performs worse than DPO-Vanilla, which seems inconsistent with the authors' claims. Is there any evidence that the user study was conducted rigorously? Can the authors confirm that the annotators responded faithfully?

- Q6. In Table 2, is the reported score based on CLIP similarity between the instruction and the generated image?

- Q7. When conditioning on a test-time user *w* who are not seen during training, how is the corresponding embedding $h_w$ obtained? If retraining the GNN is required for each new user, does this not limit the method's generalizability?

**Ethical Concerns:**

["NO or VERY MINOR ethics concerns only"]

**Final Justification:**

The authors have addressed the reviewers' concerns regarding the specifics of the GNN structure and the handling of unseen users during inference. They further strengthened their argument by adding new experiments to verify whether the proposed C-DPO effectively captures collaboration effects, as well as conducting a user study with real participants. Although the proposed method is based on DPO and employs a two-phase training process, I find its originality and novelty in proposing a preference-learning approach that accounts for diverse, non-uniform preferences across users, and in effectively designing a method to leverage collaboration among users with similar preferences.

**Limitations:**

The authors briefly address the limitation related to handling new users, which is also raised in the weaknesses part.

**Quality:**

2

**Strengths And Weaknesses:**

**Strengths**
- S1. A key consideration in preference-based model tuning is that the training data should ideally be collected from users with consistent preferences. Prior work often addressed this by statistically validating that participants shared similar tastes. In contrast, this paper acknowledges the diversity of individual preferences and extends the problem to personalized T2I editing tailored to each user's preference profile.
- S2. Leveraging GNNs to integrate information from users with similar preferences into the user embedding process is a well-motivated design choice that supports the goal of collaborative preference modeling.


**Weaknesses**
- W1. The architecture and training strategy of the GNN used for encoding user information are not clearly described. If omitted for brevity, providing a reference to relevant prior work would help readers understand the method.
- W2. Key details are missing regarding the structure and usage of the dataset, especially how preference annotations are formatted and how they are integrated with GNN-based C-DPO training. Providing concrete examples would help readers understand the overall training pipeline.
- W3. At inference time, the method for obtaining embeddings for users not seen during training is not explained. If the current approach requires re-training the GNN whenever a new user is introduced, it would significantly limit the practicality and generalizability of the method.
- W4. The evaluation of the collaborative effect is primarily based on user studies. While user feedback is valuable, more in-depth quantitative analysis is needed to verify whether correlated attributes, implicitly preferred by similar users, are actually reflected in generated images. For instance, in the case of a "rustic-style" preference mentioned by the authors, it would be helpful to evaluate quantitatively whether attributes strongly associated with rusticity are indeed realized even they are not explicitly specified.

---

> ### Author Rebuttal · Authors · 2025-07-30
>
> **W1: The architecture and training strategy of the GNN.**
>
> We thank the reviewer for highlighting this important point, and we agree that some details were briefly discussed in Section 4.1.1. due to space limitations. We would like to clarify our architecture and training strategy. Specifically, we utilize a lightweight GNN to compute contextualized user embeddings by aggregating neighbor information in our heterogeneous graph of users and attributes. Attribute node features are dense text embeddings of editing instruction topics (such as watercolor, cats, etc). The model consists of a GNN encoder of 2 GraphSAGE convolutions using mean aggregation and an edge decoder of 2 linear layers. The GNN is pretrained on an auxiliary task of predicting user–attribute links (labeled positive or negative) to ensure semantically meaningful user representations. We train on 60% of edges in our graph, withholding 20% for validation and testing each. We will elaborate this further in our final manuscript.
>
> **W2: How preference annotations are formatted and how they are integrated with GNN-based C-DPO training.**
>
> During GNN pretraining, we leverage both liked and disliked user-attribute pairs in an auxiliary edge prediction task. For instance, edges may take the form of (“user1”, “likes”, “neon green”) or (“user2”, “dislikes”, “rainbows”). Once pretraining is complete, we discard the classification head and retain the resulting user embeddings. In the full C-DPO training stage, we use these user embeddings in combination with the caption of the target image (e.g., “White ceramic bowl”). Positive and negative pairs are constructed using editing instructions that the user liked or disliked (e.g., “Add a holographic pattern to the bowl” as a positive, and “Overlay a floral pattern on the bowl” as a negative).
>
> **W3 & Q7: When conditioning on a test-time user w who is not seen during training, how is the corresponding embedding h_w obtained?**
>
> Indeed, our approach leverages an inductive embedding framework (GraphSAGE) rather than a transductive one, explicitly designed to generalize efficiently to new users without retraining. Traditional transductive methods are inherently limited to fixed graphs, unable to generalize embeddings to unseen nodes or evolving structures.  Rather than generating isolated embeddings for each user, GraphSAGE learns a set of aggregation functions which intake feature information from a node’s neighborhood and aggregates. Applying these aggregation functions efficiently generates embeddings for previously unseen users based solely on their attribute connections, provided the attribute schema matches the training data.
>
> At inference time, embedding a new user does not require retraining. Instead, a new user provides a small set of liked/disliked attributes; we then add a corresponding user node with a fixed-width, zero-initialized feature vector, link it to relevant attribute nodes, and apply our learned aggregation functions in the pretrained GNN encoder. The resulting embedding is computed inductively from the attributes and neighborhood structure alone. While periodic fine-tuning on the expanded graph may further enhance embedding quality, it is optional, not essential for inference, thus preserving the practicality and scalability of our method.
>
> **W4: (..) more in-depth quantitative analysis is needed to verify whether correlated attributes, implicitly preferred by similar users, are actually reflected in generated images.**
>
> In response to the reviewer’s suggestion, we conducted additional quantitative experiments to assess whether correlated attributes implicitly preferred by similar users are reflected in the generated outputs. For each user, we identified their top 10 most similar users based on the user-preference graph and collected the attributes those users liked. We then measured the CLIP between these collected preferences and the images generated by our method (CLIP-Neighbor), using the same evaluation set as in Tables 1 and 2 of the Appendix:
>
> | Method      | Clip-Neighbor ↑   |
> | ----------- | ----------------- |
> | SFT         | 0.286 ± 0.025     |
> | DPO-Vanilla | 0.304 ± 0.026     |
> | DPO-User    | 0.297 ± 0.026     |
> | **Ours**    | **0.312 ± 0.020** |
>
> We will include this experiment to camera ready.
>
> **Q1: How is the GNN constructed? (..) How is the computational cost?**
>
> The users and their respective liked/disliked edit instructions are represented in a heterogeneous bipartite graph, with users as one node type and attributes as another. We condense edit instructions to keyword attributes to capture broader topics. The graph is not fully connected as users are only connected to attributes they have history with. Users are only indirectly connected via shared keywords; there are no direct user–user edges. The GNN architecture (see W1 above) is pretrained on this graph and only takes ~2 minutes and is frozen for C-DPO training, which remains efficient, completing in about one hour (see Appendix Table 8 for details).
>
> For similarity metric: we rank the neighbors by the # of shared preference attributes i.e. one-hot neighbors in the graph. Thus, the nearest neighbors are the ones who like the most attributes in common with a user.
>
> **Q2: In Eq. 7, it seems that the positive/negative pairs (y+, y-) are defined based on user u's preference. (...)**
>
> The positive/negative pair (y+,y−) is defined by the anchor user u. The collaborative term does not relabel pairs using a neighbor’s preferences; instead, it re‑evaluates the same pair under the policy conditioned on a neighbor v. This encourages the policy to prefer u’s chosen edits in contexts where like‑minded neighbors would also agree, providing a graph‑weighted regularizer on top of the primary per‑user DPO term.
>
> Potential mismatch is controlled in two ways. (i) We choose only the K nearest neighbors, so the auxiliary signal comes from users with aligned tastes. (ii) The collaborative contribution is weighted by w_uv and scaled by λ ensuring it cannot overpower the individual term. This also ensures that each neighboring user gets a signal that is proportional to how similar those users are to the anchor user. Practically, this design augments u’s sparse supervision with corroborating signals from nearby users.
>
> **Q3-1: How is the dataset constructed?**
>
> Please see our answer at reviewer F3Yh-Q3.
>
> **Q3-2:  Instruction generation in a user-agnostic manner vs user u's preferences. How is p_u generated?**
>
> Instruction Generation (user-agnostic): Initially, we fine-tune an LLM using LoRA adapters in a user-agnostic manner. This model generates general-purpose editing instructions conditioned solely on image captions and high-level text prompts, without user-specific embeddings.
>
> Personalization via Preferences (user-specific): Personalized instructions p_u are generated by injecting user-specific embeddings derived from our trained GNN. Specifically, each user's embedding (obtained via GraphSAGE from their profile p_u) is converted into soft tokens, prepended to the textual input prompt, and passed through the fine-tuned LoRA-adapted LLM. This strategy yields personalized instructions that reflect the user's editing preferences.
>
> The overall format of the dataset includes pairs of base-image captions and edit instructions, annotated with explicit user identifiers and associated binary preference labels (liked or disliked). Each user profile p_u contains structured sets of preferred attributes (positive) and explicitly avoided attributes (negative), spanning several dimensions such as stylistic themes (e.g., rustic, luxury), color palettes, lighting preferences, visual textures (e.g., distressed, polished), object-level attributes (e.g., wooden beams, marble finishes), and decorative overlays (e.g., floral, cosmic effects).
>
>
> **Q4: In comparison experiments, were the prompts used for baseline methods augmented with both positive and negative attributes from P(u)?**
>
> In our comparisons, we indeed augmented the input prompts for all baseline T2I editing methods with both positive and negative attributes explicitly derived from the user's profile P(u). Specifically, we standardized prompt inputs across all tested methods by explicitly providing both liked (positive) and disliked (negative) preferences from each user’s profile. This ensured that all methods received identical, detailed conditioning information, enabling a fair and direct comparison of their personalization capabilities.
>
> **Q5: In col. 1 (Q1) of Table 1, DPO-User performs worse than DPO-Vanilla, which seems inconsistent with the authors' claims. (...)**
>
> While it is surprising that DPO-User performed slightly worse than DPO-Vanilla on Q1 in our user study, we confirm that our user study was conducted rigorously and carefully controlled. Annotators were recruited via a reputable platform (Prolific) with screening to ensure participants responded reliably. When considering quantitative metrics (CLIP similarity scores reported in Table 2), the DPO-User model outperformed DPO-Vanilla, confirming that user-specific embeddings do, in fact, better capture personalized intent.
>
> **Q6: In Table 2, is the reported score based on CLIP similarity between the instruction and the generated image?**
>
> In Table 2, the reported scores represent CLIP similarity between the generated textual instructions and the ground-truth textual user preferences. We agree that image-based metrics are also critical for evaluating our method's effectiveness. Therefore, in Tables 1, 2, and 3 of the supplementary material, we provide extensive image-based metrics. Specifically, in Table 1 (supplementary) "CLIP-T-Pref" explicitly reports CLIP similarity computed between the edited images and the user's textual preferences, directly assessing how faithfully the images align with user intent.

---

> > ### Comment · Area_Chair_MZ6e · 2025-08-05
> >
> > Dear Reviewer,
> >
> > The authors have already responded to your initial questions. As the deadline for the reviewer-author interaction session is approaching on August 6th, please begin addressing any further concerns or questions you may have. If you have no additional queries, kindly update your rating and submit your final decision.
> >
> > Thank you for your valuable contributions to NeurIPS.
> >
> > Best, AC

---

> ### Comment · Reviewer_bCfR · 2025-08-06
>
> I appreciate the authors' detailed rebuttal. The clarification helped improve my understanding of how the dataset is used to construct the GNN and how the model handles unseen users at inference time without additional training. This capability significantly contributes to the practical applicability of the proposed method. I believe that incorporating these explanations into the manuscript would enhance its clarity.
>
> I would like to ask a follow-up question. As far as I understand, the current experiments in the draft seem to evaluate the users that were also seen during training. Could the authors clarify whether this is the case? If so, the validity of the proposed method would be further strengthened by including an evaluation on unseen users through a proper train/test split of the dataset.
>
> Additionally, my concerns regarding the fairness of the comparisons with baselines have been addressed through the authors' response. It would be beneficial to include the detailed prompting conditions used for evaluating those baselines in the manuscript to support reproducibility and transparency.
>
> Lastly, the proposed method appears to follow a two-phase pipeline: i) GNN construction and training, and ii) subsequent diffusion model fine-tuning with DPO. I wonder whether it is possible to extend this framework toward an end-to-end training setup. If such an integration is feasible, I believe it would further enhance the novelty and originality of the work.

---

> ### Author Response · Authors · 2025-08-06
>
> ***Q: As far as I understand, the current experiments in the draft seem to evaluate the users that were also seen during training. Could the authors clarify whether this is the case? If so, the validity of the proposed method would be further strengthened by including an evaluation on unseen users through a proper train/test split of the dataset.***
>
>
> We clarify that our current evaluation is indeed conducted on a held-out test set of unseen users (i.e., users who were not included during training); otherwise, the evaluation would not constitute a fair assessment of our method's generalization capability. After generating the data, we explicitly divided it into training and test splits, where approximately 2900 users were included in the training set and 100 users were reserved exclusively for testing. The test users were selected to exhibit diverse preferences to ensure broad coverage of the dataset. A clear example of our test split is provided under `Supplementary Material/Code/dataset/test`. We will explicitly emphasize this important detail regarding the train/test split in the revised manuscript to ensure maximum clarity and reproducibility.
>
> ***Q: Additionally, my concerns regarding the fairness of the comparisons with baselines have been addressed through the authors' response. It would be beneficial to include the detailed prompting conditions used for evaluating those baselines in the manuscript to support reproducibility and transparency.***
>
> We thank the reviewer for highlighting this aspect. We agree that explicitly documenting these conditions is important for reproducibility and transparency. As an illustrative example, when evaluating baselines with combined liked and disliked preferences, we used this structured prompt:
>
> ~~~
> System Prompt:
> "You are an expert at providing user personalized image-editing instructions."
>
> User Prompt:
> "Likes: [e.g. colorful and playful styles, unicorn motifs, rainbows]
> Dislikes: [e.g. minimalistic styles, muted tones, overly realistic imagery]
>
> Original Image Caption: [e.g. "A white ceramic bowl"]
>
> Suggest a single, cohesive image-editing instruction."
> ~~~
>
>
> We use the same prompt template across all baselines to ensure fairness during evaluation. In our revised manuscript, we will explicitly include this detailed prompting template, clearly specifying how user preferences (liked, disliked, or both) were integrated.
>
> ***Q: Lastly, the proposed method appears to follow a two-phase pipeline: i) GNN construction and training, and ii) subsequent diffusion model fine-tuning with DPO. I wonder whether it is possible to extend this framework toward an end-to-end training setup. If such an integration is feasible, I believe it would further enhance the novelty and originality of the work.***
>
> We share the reviewer’s enthusiasm for an end-to-end alternative and thank them for highlighting this direction. Our current two-phase procedure is motivated by practical stability and we would like to clarify the role of each phase. Phase 1 serves two critical purposes: (i) a lightweight link-prediction pre-training that prevents cold-start collapse of user embeddings, and (ii) supervised fine tuning of a lightweight LoRA adapter that becomes the frozen reference, an essential step that supplies a well-formed editing-instruction providing as stable anchor for the KL term during phase 2.
>
> Phase 2 then updates the GNN, the projection MLP (soft-prompt tokens), and a trainable LoRA policy adapter jointly under the collaborative DPO loss; thus every parameter that learns from user preferences is trained end-to-end once phase 2 begins.
>
> Nonetheless, integrating these components into a single phase unified, end-to-end training framework represents an exciting direction for future exploration and we will clearly discuss this opportunity in the revised manuscript.

---

> ### Comment · Reviewer_bCfR · 2025-08-07
>
> Thank you for the detailed response. My concerns regarding the experimental details have been resolved, and the practicality of the proposed method has been well demonstrated, particularly through its ability to handle new users at inference time.
>
> As Reviewer bMep pointed out, conducting experiments using real human preference data, rather than synthetic data, could further strengthen the empirical validity and usefulness of the approach.

---

> > ### Author Response · Authors · 2025-08-07
> >
> > We sincerely thank the reviewer for their positive feedback and for acknowledging our efforts to address the previous concerns. We fully agree that evaluating our method with real human preference data would significantly strengthen the empirical validity.
> >
> > Motivated by the earlier suggestions from Reviewer bMep, we have run a user study on real user preference data. For detailed information about our real-user experiments, please kindly refer to the section marked by the keyword `Q1: Personalization` on this page.

---

### Note · Authors · 2025-08-12

We thank all reviewers and the Area Chair for their constructive feedback and engaging discussions throughout the rebuttal period. We have addressed ***40 questions from 4 reviewers*** throughout the rebuttal period, and provided new analyses and comparisons which will be incorporated into the revised manuscript, including:

- ***Two new user studies with real-preference data*** validating C-DPO’s effectiveness beyond synthetic personas. Results showed that users rated our edits as highly personalized and disentangled, and consistently preferred them over baselines.

- We explicitly evaluated whether our framework transfers implicit attributes favored by like-minded users into a target user’s outputs (see CLIP-Neighbor experiments), confirming it ***successfully incorporates neighbor-inferred preferences without explicit prompting.***

- We performed requested prompt-engineering baselines to test if sophisticated textual injection of user and neighbor preferences could match GNN-based collaboration. Our results showed that ***our structured embeddings yield richer personalization than raw text prompts.***

- We compared C-DPO against state-of-the-art instruction-based and unified editing models (BAGEL, SEED-LLaMA, Nexus-Gen) to assess whether personalization gains persist against the stronger methods. C-DPO ***consistently outperformed on CLIP-I-PREF, DINO, and HPS metrics, validating its superiority even in competitive settings.***

- We ran an ***ablation on the number of nearest neighbors (K)***  and analyzed ***sensitivity of the collaborative term λ***.

- We ***confirmed baseline fairness*** by ensuring all methods received identical positive and negative preference attributes in prompts via a standardized template.

- ***Clarifications on key methodological details***, including the GNN input design during training and inductive inference for unseen users, the full prompting conditions used to ensure reproducibility, and the synthetic dataset generation pipeline with attribute curation.

Our work introduces the `first framework for personalized image editing in T2I diffusion models`, with a novel graph-regularized collaborative term that fuses explicit user preferences with implicit attributes from like-minded users, ***a functionality absent in prior methods***. We believe our method’s technical advances along with establishing a new problem setting for generative AI community make it a timely and valuable contribution for the NeurIPS community.

---

### Decision · Program_Chairs · 2025-09-17

**Decision:**

Accept (poster)

**Comment:**

The paper introduces the first framework for personalized image editing in diffusion models. During the rebuttal, the authors addressed the reviewers’ initial questions thoroughly, by including new baseline comparisons and user studies. Three out of four reviewers were satisfied with the responses and recommended acceptance.

One reviewer, however, remained unconvinced, arguing that the techniques were similar to federated learning. The AC considers this criticism unpersuasive and agrees with the authors that reusing techniques from other fields does not diminish the novelty if they are effectively applied to a new problem. The authors also made strong contributions by integrating GNNs and DPO into diffusion models.

Overall, the paper proposes a timely and interesting problem, presents a clever solution for personalized diffusion models, and opens up a promising research direction in personalized generative AI with potential long-lasting impact. The AC recommends acceptance (poster).

The authors are encouraged to incorporate the new experiments and discussions from the rebuttal into the final version.